# Automatic and feature-specific prediction-related neural activity in the human auditory system

Gianpaolo Demarchi [1,3], Gaëtan Sanchez [1,2,3] & Nathan Weisz[1]

Prior experience enables the formation of expectations of upcoming sensory events. However, in the auditory modality, it is not known whether prediction-related neural signals carry feature-specific information. Here, using magnetoencephalography (MEG), we examined whether predictions of future auditory stimuli carry tonotopic specific information. Participants passively listened to sound sequences of four carrier frequencies (tones) with a fixed presentation rate, ensuring strong temporal expectations of when the next stimulus would occur. Expectation of which frequency would occur was parametrically modulated across the sequences, and sounds were occasionally omitted. We show that increasing the regularity of the sequence boosts carrier-frequency-specific neural activity patterns during both the anticipatory and omission periods, indicating that prediction-related neural activity is indeed feature-specific. Our results illustrate that even without bottom-up input, auditory predictions can activate tonotopically specific templates.

[1] Centre for Cognitive Neuroscience and Division of Physiological Psychology, University of Salzburg, Hellbrunnerstraße 34, 5020 Salzburg, Austria. [2] Lyon Neuroscience Research Center, Brain Dynamics and Cognition Team, INSERM UMRS 1028, CNRS UMR 5292, Université Claude Bernard Lyon 1, Université de Lyon, F-69000 Lyon, France. [3]These authors contributed equally: Gianpaolo Demarchi, Gaëtan Sanchez. Correspondence and requests for materials should be addressed to G.D. (email: gianpaolo.demarchi@sbg.ac.at)

The human capacity to predict incoming sensory inputs based on past experiences is fundamental to our ability to adapt our behavior to complex environments. A core enabling process is the identification of statistical regularities in sensory input, which does not require any voluntary allocation of processing resources (e.g., selective attention) and occurs more or less automatically in healthy brains[1]. Analogous to other sensory modalities[2,3], auditory cortical information processing takes place in hierarchically organized streams along putative ventral and dorsal pathways[4]. These streams reciprocally connect different portions of auditory cortex with frontal and parietal regions[4,5]. This hierarchical anatomical architecture yields auditory cortical processing regions sensitive to top-down modulations, thereby enabling modulatory effects of predictions. Building upon this integrated feedforward and top-down architecture, cortical and subcortical regions seem to be involved in auditory prediction-error generation mechanisms[6,7]. A relevant question in this context is to what extent prediction-related top-down modulations (pre-)activate the same feature-specific neural ensembles as established for genuine sensory stimulation.

Such fine-tuning of neural activity is suggested by frameworks that propose the existence of internal generative models[8–11], inferring the causal structure of sensory events in our environment and the sensory consequences of our actions. A relevant process to validate and optimize these internal models is to predict incoming stimulus events by influencing the activity of corresponding neural ensembles in relevant sensory areas. Deviations from these predictions putatively lead to (prediction) error signals, which are passed on in a bottom-up manner to adapt the internal model, thereby continuously improving predictions[9] (for an alternative predictive coding architecture see[12]). According to this line of reasoning, predicted input should lead to weaker neural activation than input that was not predicted, which has been illustrated previously in the visual[13] and auditory modality[14]. Support for the idea that predictions engage neurons specifically tuned to (expected) stimulus features has been more challenging to address and has come mainly from the visual modality (for review see ref. [15]). In an fMRI study, Smith and Muckli showed that early visual cortical regions (V1 and V2), which process occluded parts of a scene, carry sufficient information to decode above-chance different visual scenes[16]. Intriguingly, activity patterns in the occlusion condition are generalized to a non-occlusion control condition, implying context-related top-down feedback or input via lateral connections to modulate the visual cortex in a feature-specific manner. In line with these results, it has been shown that mental preplay of a visual stimulus sequence is accompanied by V1 activity that resembles activity patterns driven in a feedforward manner by the real sequence[1]. Beyond more or less automatically generated predictions, explicit attentional focus to specific visual stimulus categories also goes along with similar feature-specific modifications in early and higher visual cortices even in the absence of visual stimulation[17]. It has been proposed that expectations increase baseline activity of sensory neurons tuned to a specific stimulus[18,19]. Moreover, another study using magnetoencephalography (MEG) and multivariate decoding analyses revealed how expectation can induce a preactivation of stimulus template in visual sensory cortex, suggesting a mechanism for anticipatory predictive perception[20]. Overall, for the visual modality, these studies emphasise that top-down processes lead to sharper tuning of neural activity to contain more information about the predicted and/or attended stimulus (feature).

Studies that look at whether predictions in the auditory domain (pre-)activate specific sensory representations in a sharply tuned manner are scarce especially in humans (for animal works see e.g., refs. [21,22].). Sharpened tuning curves of neurons in A1 during selective auditory attention have been established in animal experiments[23], although this does not necessarily generalize to automatically formed predictions. A line of evidence could be drawn from research in marmoset monkeys, in which a reduction of auditory activity is seen during vocalization[24] (for suppression of neural activity to movement-related sounds in rats in rats see ref. [25]). This effect disappears when fed back vocal utterances are pitch shifted[26], thereby violating predictions. Such an action-based dynamic (i.e., adaptive) sensory filter likely involves motor cortical inputs to the auditory cortex that selectively suppress predictable acoustic consequences of movement[27]. Interestingly, even inner speech may be sufficient to produce reduced neural activity, but only when the presented sounds match those internally verbalized[28]. A study using invasive recordings in a small set of human epilepsy patients showed that masked speech is restored by specific activity patterns in bilateral auditory cortices[29], an effect reminiscent of a report in the visual modality[1] (for other studies investigating similar auditory continuity illusion phenomena see refs. [30–32]). Although feature specific, this "filling-in" type of activity pattern observed during phoneme restoration cannot conclusively determine whether this mechanism requires top-down input. In principle, these results could also be largely generated via bottom-up thalamocortical input driving feature-relevant neural ensembles via lateral or feedforward connections. To resolve this issue, putative feature-specific predictions need to be shown also without confounding feedforward input (i.e., during silence).

In high-resolution fMRI experiment, it was recently shown that predictive responses to omissions follow a tonotopic organization in the auditory cortex[33]. But following the notion that predictive processes are also proactive in an anticipatory sense, the exact timing of the effects provides important evidence on whether predictions in the auditory system occur together with feature-specific preactivations of relevant neural ensembles[15]. To this end, higher temporal resolution techniques (e.g., electro-encephalography (EEG) or MEG) are needed.

The goal of the present MEG study was to investigate in healthy human participants whether predictions in the auditory modality are exerted in a carrier-frequency (i.e., tonotopic) specific manner and, more importantly, whether those predictions are accompanied by anticipatory effects. For this purpose, we merged an omission paradigm with a regularity modulation paradigm (for an overview see ref. [34]; see Fig. 1 for details). So-called omission responses occur when an expected tone is replaced by silence. his response has been frequently investigated in the context of Mismatch Negativity (MMN[35]) paradigms, which undoubtedly have been the most common approach of studying the processing of statistical regularities in human auditory processing[36–39]. This evoked response occurs upon a deviation from a "standard" stimulus sequence, that is, a sequence characterized by a regularity violation regarding stimulation order. For omission responses (e.g, ref. [40]), this order is usually established in a temporal sense, allowing precise predictions of when a tone will occur[41] (for a study using a repetition suppression design see ref. [42]). The neural responses during these silent periods are of outstanding interest as they cannot be explained by any feedforward propagation of activity elicited by a physical stimulus. Thus, omission of an acoustic stimulation will lead to a neural response, as long as this omission violates a regular sequence of acoustic stimuli, that is, when it occurs unexpectedly. Previous works have identified auditory cortical contributions to the omission response (e.g., ref. [42].). Interestingly, and underlining the importance of a top-down input driving the omission response, a recent DCM study by Chennu et al.[43] illustrates that it can be best explained when assuming top-down driving inputs in higher-order cortical areas (e.g.,

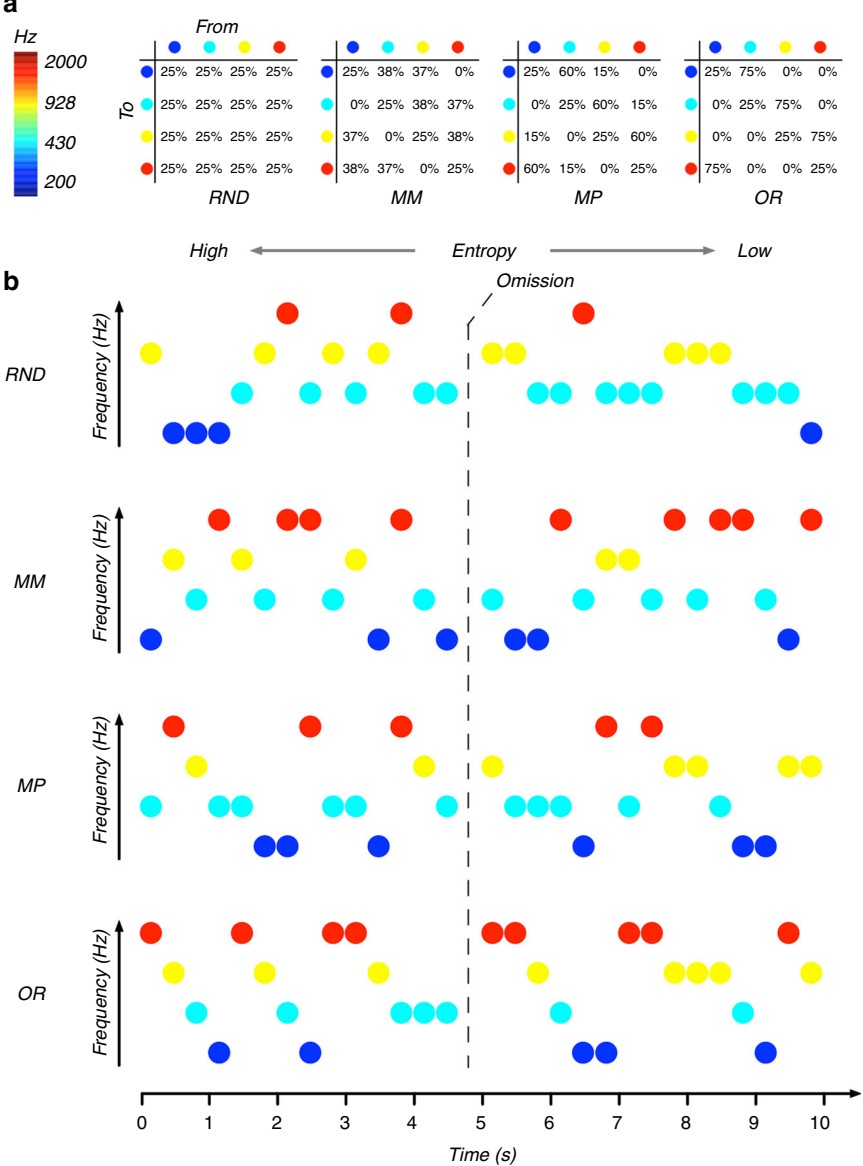

**Fig. 1** Experimental design. **a** Transition matrices used to generate sound sequences according to the different conditions (random (RD), midminus (MM), midplus (MP), and ordered (OR)). **b** Schematic examples of different sound sequences generated across time. 10% of sound stimuli were randomly replaced by omission trials (absence of sound) in each context

frontal cortex). While establishing temporal predictions via a constant stimulation rate, we varied the regularity of the sound sequence by parametrically modulating its entropy level (see e.g., refs. [44,45].). Using different carrier frequencies, sound sequences varied between random (high entropy; transition probabilities from one sound to all others at chance level) and ordered (low entropy; transition probability from one sound to another one above chance). Our reasoning was that omission-related neural responses should contain carrier-frequency specific information that is modulated by the entropy level of the contextual sound sequence. Using a time generalization decoding approach[46], we find strong evidence that particularly during the low entropy (i.e., highly ordered) sequence neural activity prior to and during the omission period contains carrier-frequency specific information similar to activity observed during real sound presentation. Our work reveals how even in passive listening situations, the auditory system extracts the statistical regularities of the sound input, continuously casting feature-specific (anticipatory) predictions as (pre-)activations of carrier-frequency specific neural ensembles.

## Results

**Single-trial neural activity contains carrier frequency information.** A crucial first step and the foundation to addressing the question of whether carrier-frequency specific neural activity patterns are modulated by predictions even in anticipatory or omission periods is to establish that we can actually decode carrier-frequencies during actual sound presentation. To address this issue, we used the single trial MEG time-series data from the random (high entropy) condition and trained a classifier (LDA) to distinguish the four different carrier frequencies (Fig. 2a). Robust above chance ($p < .05$, Bonferroni corrected, gray line) classification accuracy was observed commencing ~30 ms following stimulus onset and peaking (~35%) at around 100 ms to gradually decline until 350 ms. Interestingly, carrier-frequency specific information remained above chance at minimum until 700 ms post-stimulus onset (i.e., the entire period tested) meaning that this information was contained in the neural data when new sounds were presented. In this study, we focus on the data from the random sequence, since this decoding was the basis (i.e., training data set) for all upcoming

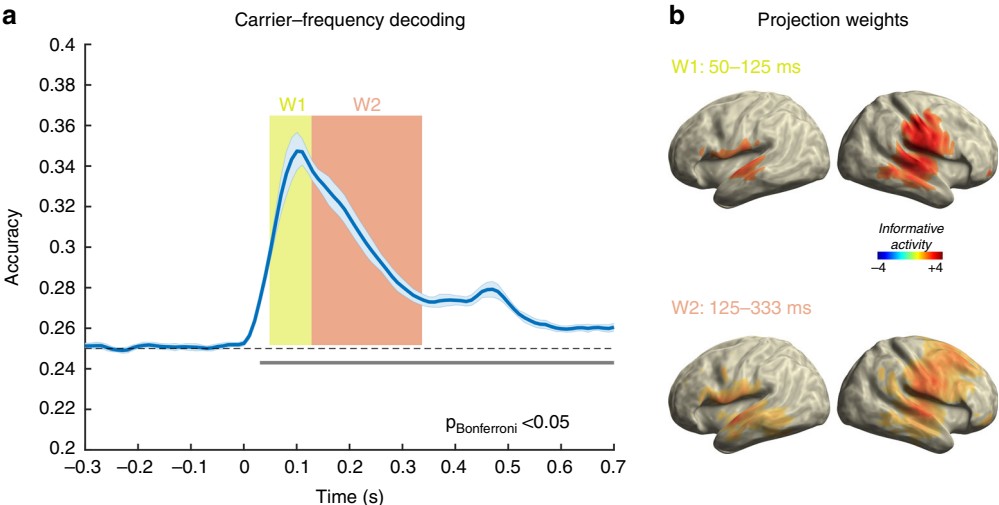

**Fig. 2** Decoding carrier frequencies from random sound sequences. **a** Robust increase of decoding accuracy is obtained rapidly, peaking ~100 ms after which it slowly wanes. Note however that significant decoding accuracy is observed even after 700 ms, i.e. likely representing a memory trace that is (re-) activated even when new tones (with other carrier frequencies) are processed. **b** Source projection of classifier weights (relative change baseline (-100-0) ms, 50% threshold) for an early (W1) and late (W2) reveals informative activity to mainly originate from auditory cortices, with a right hemispheric dominance. During the later (W2) period informative activity spreads to also encompass e.g. frontal regions

analysis using a time-generalization approach, in which carrier-frequency had to be decoded.

To identify the brain regions that provided informative activity, we adapted a previously reported approach[47] that projects the classifier weights obtained for the decoding of carrier frequency from sensor to source space (Fig. 2b). Since later analysis using the time-generalization approach pointed to a differential entropy-level effect for early (50–125 ms; W1) and late (125–333 ms; W2) periods of the training time period (described below and in Fig. 3), the projected weights are displayed separately for these periods. For both time periods it is evident that bilateral auditory cortical regions contribute informative activity, albeit with a right hemispheric dominance. While this appears similar for both time-windows, informative activity also spreads to non-auditory cortical regions such as frontal areas during the later (W2) period.

Overall, the analyses so far show that carrier-frequency specific information can be robustly decoded from MEG, with informative activity originating (as expected) mainly from the auditory cortex. Interestingly and beyond what can be shown by conventional evoked-response analysis, carrier-frequency specific information is temporally persistent, potentially reflecting a memory trace of the sound that is (re-)activated when new information arrives.

**Modulations of entropy lead to clear anticipatory prediction effects.** An anticipatory effect of predictions should be seen as relative increases of carrier-frequency specific information prior to the onset of the expected sound with increasing regularity of the sound sequence. To tackle this issue it is important to avoid potential carry-over effects of decoding, which would be present, for example, when training and testing on ordered sound sequences. We circumvented this problem by training our classifier only on data from the random sound sequence (see Fig. 2a) and testing it for carrier-frequency specific information in all other conditions using time generalization. The raw (i.e., grand averaged) condition- and time-generalized results are displayed in Fig. 3a, b, and entropy-dependent modulation of decoding accuracy in the pre-sound (Fig. 3c) and pre-omission period (Fig. 3d) can be readily appreciated. A non-parametric cluster permutation test (Fig. 3c, d) yields

a clear prestimulus effect in both cases confirming a linear increase of decoding accuracy with a decreasing entropy level (pre-sound: $p_{cluster} < 0.001$; pre-omissions: $p_{cluster} < 0.001$). In both cases, these anticipatory effects appear to involve features that are relevant at later training time periods (~125–333 ms, W2; for informative activity in source space see also Fig. 2b). The time courses of averaged accuracy for this training time interval (shown in Fig. 3e: analysis locked to sounds, Fig. 3f: analysis locked to omissions) visualize this effect with a clear relative increase of decoding accuracy prior to expected sound onset in particular for the ordered sequence. This analysis clearly underlines that predictions that evolve during a regular sound sequence contain carrier-frequency specific information that are preactivated in an anticipatory manner, akin to the prestimulus stimulus templates reported in the visual modality[20].

**Entropy-dependent classification accuracy of sounds.** According to some predictive processing frameworks[15], predicted sounds should lead to a reduced activation as compared with cases in which the sound was not predicted. In the case that the activation stems mainly from carrier-frequency specific neural ensembles, a decrease of decoding accuracy with increasing regularity (i.e., low entropy) could be expected. Using a time generalization approach, we applied the classifier trained on the carrier-frequencies from the random sound sequence (Fig. 2) to the post-sound periods of the individual entropy conditions (see grand-average time- and condition generalization results in Fig. 3a). Our regression approach yielded a negative relationship between decoding accuracy and regularity at ~100–200 ms post-sound onset; however, this effect did not survive multiple comparison testing ($p_{cluster} = 0.12$) (Fig. 3c). Also on a descriptive level, this effect appeared not to be strictly linear, meaning that our study does not provide strong evidence that predictions reduce carrier-frequency specific information of expected tones. Interestingly, a significant positive association ($p_{cluster} < 0.001$) at a later interval time period emerged, beginning at around 370 ms post-sound onset and lasting at minimum until 700 ms (Fig. 3c). Analogous to the aforementioned anticipation effect (and the subsequently described omission effect), decoding accuracy increased the more regular the sound sequence. However, this

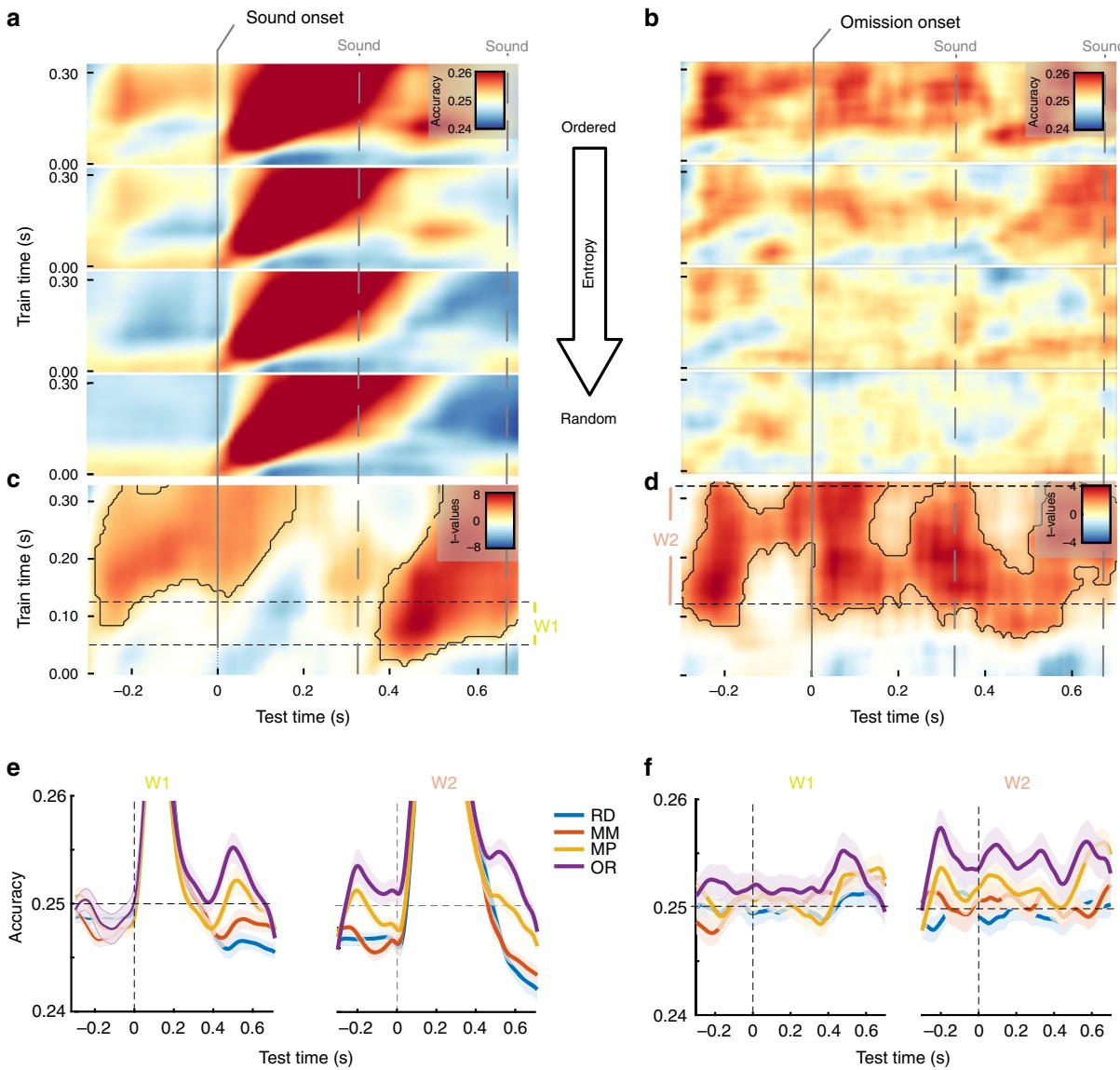

**Fig. 3** Analysis for pre- and post-stimulus decoding using time-generalization of classifier trained on random sound sequences. **a, b** "Raw" decoding time-generalization maps (grand average across subjects), tested on sound- (**a**) and omission- (**b**) locked trials, in increasing entropy from top to bottom. **c, d** Regression results (sound left, omission right), using entropy level as the independent variable; red colors indicate increased decoding accuracy for more regular sequences. $t$-values are thresholded at uncorrected $p < 0.05$. The areas framed in black are clusters significant at $p_{cluster} < 0.05$. **e, f** Decoding accuracy for individual conditions averaged for training times between the dashed lines, testing on sound (left) and omission (right). **c** Display of effects pre- and post-sound, showing a clear anticipation effect and a late effect commencing after ~400 ms. The latter effect is more clearly visualized in (**e**). Interestingly, different train times appear to dominate the anticipation and post-stimulus effects. **d** Display of effects pre- and post-omission, showing a single continuous positive cluster. However, the actual $t$-values suggest temporally distinct maxima within this cluster underlining the dynamics around this event. Analogous to sounds a clear anticipation effect can be observed, driven by increased pre-omission decoding accuracy for events embedded in regular sequences (see **f**). A similar increase can be seen immediately following the onset of the omission which cannot be observed following actual sound onset. Interestingly this increase is long lasting with further peaks emerging approximately at 330 ms and 580 ms

effect was most strongly driven by classifier information stemming from earlier training time intervals (50–125 ms, W1; see Figs. 2b and 3e, left panel). This effect is in line with the previously described temporally extended effect for the decoding of carrier frequency from random sound sequences and suggests that carrier-frequency specific information is more strongly reactivated by subsequent sounds when embedded in a more regular sound sequence.

**Entropy-dependent classification accuracy of sound omissions.**
Our sound sequences were designed such that the onset of a sound could be precisely predicted (following the invariant 3 Hz

rhythm), but the predictability of the carrier-frequencies was parametrically modulated according to the entropy level. Following the illustration of anticipation-related prediction effects, a core question of our study was whether we could identify carrier-frequency specific information following an expected but omitted sound onset. This is suggested by the grand-average time- and condition-generalized decoding results time-locked to the omission onset shown in Fig. 3b. A nonparametric cluster permutation test for the regression (Fig. 3d) yielded a single cluster that also comprises the anticipation effect described above ($p_{cluster} < 0.001$); however—and in contrast to the analysis locked to sounds (Fig. 3c)—a post-omission onset increase is clearly identifiable.

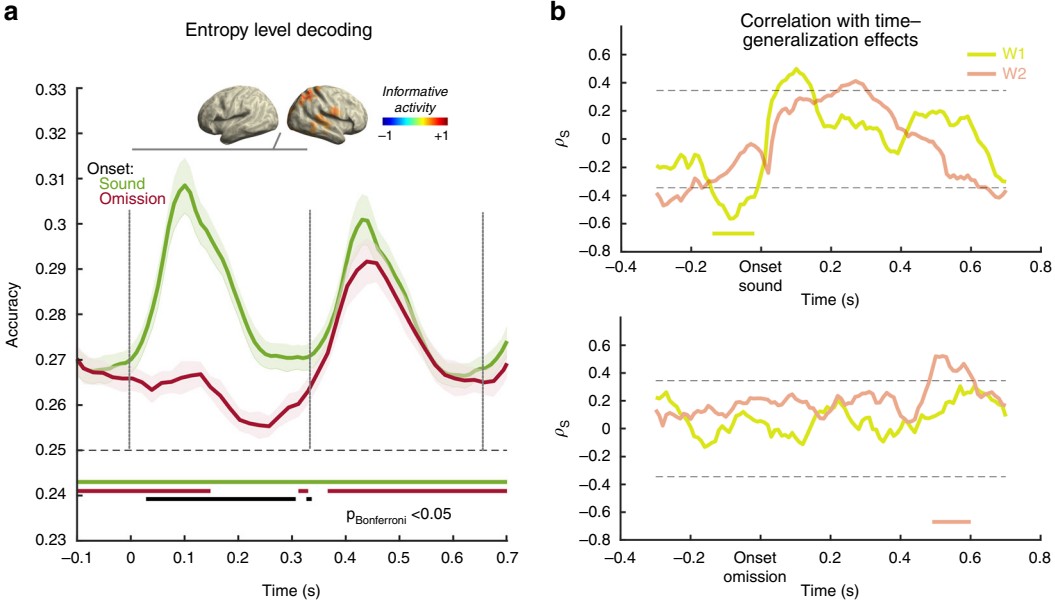

**Fig. 4** Decoding entropy level of sound sequence and correlation main prediction effects gained from time-generalization analysis (see Fig. 3). **a** A classifier was trained to decode the entropy level from MEG activity elicited by sounds and tested around relevant events, i.e. sound or omission onset. Robust increases of decoding accuracy can be observed following sound onsets. Right temporal and parietal regions appear to contribute most informative activity (small inset, relative change baseline (-100-0 ms), 50% threshold). While overall decoding accuracy is above chance level throughout most of the entire period, this pattern breaks down briefly following an omission. **b** Average entropy level decoding following sound onset (0–330 ms) was taken and (Spearman) correlated with the time-generalized decoding accuracy of the low entropy condition. Nonparametric cluster permutation test yields a significant negative correlation especially with early training time-window patterns (W1) in the anticipation period towards a sound that was; however, not observed prior to omissions. Following the onset of omissions nonparametric cluster permutation testing pointed to a late positive correlation with the late activation patterns (W2)

Interestingly, also the post-omission effect was long lasting, reaching far beyond the short interval of the omitted sound. Figure 3f shows the decoding accuracy averaged over the W1 and W2 training time intervals (see Fig. 2b), illustrating the enhanced decoding accuracy especially for the ordered sequence particularly pronounced for W2. Even though the entropy-driven effect is clearly continuous, local peaks at ~90, 330, and 580 ms following the time of the (omitted) stimulation onsets can be identified at a descriptive level. This shows that carrier-frequency specific information about an anticipated sound is enduring, not only encompassing prestimulus periods, but temporally persists when the prediction of a sound presentation is violated.

**Interindividual neural representations of statistical regularities and their correlation with feature-specific predictions.** Our previous analyses established a clear relationship between (anticipatory) predictions of sounds and the carrier-frequency specific information contained in the neural signal. This was derived by a regression analysis across conditions using the entropy level as independent variable. An interesting follow-up albeit exploratory question is whether interindividual variability to derive the statistical regularity from the sound sequences would be correlated with carrier-frequency specific information for predicted sounds.

To pursue this question we first tested whether information pertaining to the level of statistical regularity of the sequence is contained in the single-trial level signal. Using all magnetometers (i.e., discarding the spatial pattern) and a temporal decoding approach showed that the entropy level of the condition in which the sound was embedded could be decoded above chance from virtually any time point (Fig. 4a, green curve). Given the block-wise presentation of the conditions, this temporally extended

effect ($p < 0.05$, Bonferroni corrected, green horizontal line) is not surprising. On top of this effect, a transient increase of decoding accuracy following ~50–200 ms stimulus onset can be observed. In order to identify potential neural generators that drive the described effect, the trained classifier weights were projected in the source space (see inset of Fig. 4a) analogous to the approach described above. Since sensor level analysis suggested a temporally stable (in the sense of almost always significant) neural pattern, the entire 0–330 ms time period was used for this purpose (gray line, inset of Fig. 4a). The peak informative activity was strongly right lateralized to temporal as well as parietal regions. Based on this analysis, we can state that information about the regularity of the sound sequence is contained also at the single-trial level and that temporal and parietal region may play an important role in this process. We applied the classifier trained on the sound presentations to the same omission periods, in order to uncover whether the statistical pattern information is also present following an unexpected omission (Fig. 4a, red curve). Interestingly, in this case, a decrease of decoding accuracy was observed commencing ~120 ms after omission onset and lasting for ~200 ms. During this brief time period the entropy level could not be decoded above chance (Fig. 4a, red horizontal line). This effect illustrates that an unexpected omission transiently interrupts the processing of the statistical regularity of the sound sequence. We also tested for a difference in accuracy between sound and omission trials at each timepoint, which was significant ($p < 0.05$, Bonferroni corrected, black horizontal line) between ~30 ms and ~300 ms, again emphasizing the brevity of this effect.

To test whether the interindividual variability in neurally representing the entropy level is associated with carrier-frequency specific predictions, we correlated average entropy decoding accuracy in a 0–330 ms time window following sound onset with

time-generalized decoding accuracy for carrier-frequency around sound or omission onset separately for the early (W1) and late (W2) training time-windows. A nonparametric cluster permutation test (Fig. 4b) shows for the early training time-window (W1) a negative relationship ($p_{cluster} = 0.02$), meaning that participants whose neural activity suggested a stronger representation of the level of statistical regularity preactivate carrier-frequency specific neural patterns to a lesser extent. It should be noted that the overall entropy-related effect was driven more by the later training-time window (W2), for which no correlation was observed in the present analysis. Also, the correlation effect was not found when locking the analysis to the omission onset, which could either suggest a spurious finding or suboptimal power for this effect, given the much lower number of trials for the omission-centered time-generalized decoding. On a descriptive (uncorrected) level positive correlations can be seen following sound onset that are sequentially pronounced for early (W1) and late (W2) training-time windows ($p_{cluster} = 0.062$ and $p_{cluster} = 0.096$, respectively). Following the omission onset a late positive correlation ($p_{cluster} = 0.033$) was observed at ~500–600 ms for the late training time-window (W2), meaning that the carrier-frequency specific pattern of the omitted sound was reactivated more during the ordered sequence for participants who showed stronger encoding of the statistical regularity. Altogether, this analysis demonstrates that the brain has a continuous representation of the magnitude of regularity of the sequence and that it is modulated by the presence or (unexpected) absence of a sound. Furthermore, some indications are present that suggest the interindividual variability in this more global representation of the input regularity could potentially influence the exertion of carrier-frequency specific neural patterns preceding and following sound, even though this connection would need to be followed up in more targeted studies.

## Discussion

In this study, we investigate neural activity during passive listening to auditory tone sequences by manipulating respective entropy levels and thereby the predictability of an upcoming sound. We used multivariate pattern analysis (MVPA) applied to MEG data to first show that neural responses contain sufficient information to decode the carrier-frequency of tones. Using classifiers trained on random sound sequences in a condition- and time-generalized manner, our main finding is that carrier-frequency specific information increases the more regular (i.e., predictable) the sound sequence becomes, especially in the anticipatory and post-omission periods. This study provides strong support that prediction-related processes in the human auditory system are sharply tuned to contain tonotopically specific information. While the finding of sharp tuning of neural activity is not surprising, given in particular invasive recordings from the animal auditory cortex (e.g., during vocalizations, see ref. [26]; or shift of tuning curves following explicit manipulations of attention to specific tone frequencies, see refs. [23,48]), our work is a critical extension of previous human studies for which tonotopically tuned effects of predictions has not been shown thus far. Critically, given that omission responses have been considered as pure prediction signals[34,49], our work illustrates that sharp tuning via predictions does not require bottom-up thalamocortical drive.

To pursue our main research question, that is whether single-trial MEG data contains carrier-frequency specific information following sound onset, we relied on MVPA applied to MEG data[46,50]. Prior to addressing whether (anticipatory) prediction-related neural activity contains carrier-frequency specific information, an important sanity check was first to establish the

decoding performance when a sound was presented in the random sequence. A priori, this is not a trivial undertaking given that the small spatial extent of the auditory cortex[51] likely produces highly correlated topographical patterns for different pure tones and that mapping tonotopic organization using noninvasive electrophysiological tools has had mixed success (for critical overview see e.g., ref. [52].). Considering this challenging background it is remarkable that all participants showed a stable pattern with marked post-stimulus onset decoding increases after ~50 ms. While a peak is reached around 100 ms post-sound onset after which decoding accuracy slowly declines, it remains above chance for at least 700 ms. This observation is remarkable given the passive setting for the participant (i.e., no task involved with regards to sound input) and the very transient nature of evoked responses to sounds that are commonly used in cognitive neuroscience. Our analysis shows that neural activity patterns containing carrier-frequency specific information remain present for an extended time putatively representing a memory trace of the sound that is available when new acoustic information impinges on the auditory system. This capacity is of key importance for forming associations across events, thereby enabling the encoding of the statistical regularity of the input stream[53,54].

The previous result underlines that noninvasive electrophysiological methods such as MEG can be used to decode low-level auditory features such as the carrier-frequency of tones. This corroborates and extends findings from the visual modality for which successful decoding of low-level stimulus features such as contrast edge orientation have been demonstrated previously[55]. However, the analysis leading to this conclusion included all sensors and was, therefore, spatially agnostic. We used an approach introduced by Marti and Dehaene[47] to describe informative activity at the source level. Based on the subsequent entropy-related effects identified in the time-generalization approach, we focussed on an earlier (W1, 50–125 ms) and a later time-window (W2, 125–333 ms). While informative activity was observed bilaterally especially in auditory regions along the Superior Temporal Gyrus in both hemispheres, the pattern was stronger on the right side. Furthermore, on a descriptive level, informative activity was spread more frontally in the later time-window, implying an involvement of hierarchically downstream regions. Overall this analysis suggests that carrier-frequency specific information mainly originates from within auditory cortical regions, but that regions not classically considered as auditory processing regions may contain feature-specific information as well[5].

This analysis was relevant not only as a sanity check, but also because the trained classifiers were used and time-generalized to all entropy levels. While this approach yielded highly significant effects (see also below for discussion), decoding accuracy was not high in absolute terms, especially for the anticipatory and omission periods. However, it should be noted that we refrained from a widespread practice of sub-averaging trials[50,55], which boosts classification accuracies significantly. When compared with cognitive neuroscientific M/EEG studies that perform decoding on the genuine single trials and focus on group-level effects (rather than feature-optimizing on an individual level as in BCI applications), the strength of our effects are comparable (e.g., refs. [47,56]).

Using an MVPA approach with time-generalization allowed us to assess whether carrier-frequency related neural activity prior to or during sound/omission is systematically modulated by the entropy level. When training and testing within regular (i.e., low entropy) sound sequences, carry-over effects could be artificially introduced that were erroneously interpreted as anticipation effects: in these conditions the preceding sound (with its carrier-frequency) already contains information about the upcoming

sound. To circumvent this problem, we consistently used a classifier trained on sounds from the random sequence, that is, where neural activity following a sound is not predictive of an upcoming sound, and applied it to all conditions. Using a regression analysis, we could derive in a time-generalized manner the extent to which carrier-frequency specific decoding accuracy of the (omitted) sound was modulated by the entropy of the sound sequence that the event was embedded in. The act of predicting usually contains a notion of pre-activating relevant neural ensembles, a pattern that has been previously illustrated in the visual modality (e.g., ref. [1,20].). For the omission response, this was put forward by Bendixen et al.[49], even though the reported evoked response effects cannot be directly seen as signatures of preactivation. Chouiter et al.[57] also use a decoding approach, and show an effect of frequency/pitch after expected but omitted sounds, but do not look for an anticipatory effect. In line with this preactivation view, our main observation was that carrier-frequency specific information increased with the increased regularity of the tone sequence already in the pre-sound/omission period, clearly showing sharp tuning of neural activity in anticipation of the expected sound. This effect was particularly pronounced for later training time periods (W2; see Fig. 3) which contained informative activity also beyond auditory regions (e.g., frontal cortex; see Fig. 2b). This finding critically extends the aforementioned research already completed in the visual modality, that clearly established feature-specific tuning of anticipatory prediction processes in the auditory system. Our finding supports and enhances a recent high field fMRI study for the auditory domain[33], where the lower temporal resolution of the technique could not permit the separation of pure prediction and and preactivation effects.

According to most predictive coding accounts, expected stimuli should lead to attenuated neural responses, which has been confirmed for auditory processing using evoked M/EEG (e.g., ref. [14]) or BOLD responses (e.g., ref. [13]). Thus a reduction of carrier-frequency specific information could also be expected when sounds were embedded in a more ordered sound sequence. While such an association was descriptively observed in early training- (W1) and testing-time (<200 ms) intervals, it was statistically not significant (Fig. 3c). This observation may be reconciled when dissociating the strength of neural responses (e.g., averaged evoked M/EEG or BOLD response) from the feature specific information in the signal, as has been described for the visual modality: here reduced neural responses in the visual cortex have been reported, while at the same time representational information is enhanced[19]. An enhancement of representational—that is, carrier-frequency specific—information was observed in our study at late testing-time intervals, following ~500 ms after sound onset. This effect was broad in terms of the training-time window; however, it was largest for early intervals (W1; see Fig. 3c). The late onset suggests this effect is a consequence of the subsequent sound presentation, that is, a reactivation of the carrier-frequency specific information within more ordered sound sequences, which would be crucial in establishing and maintaining neural representations of the statistical regularity of the sound sequence. While this effect derived from time-generalization analysis is not identical to the temporal decoding result described above, it is fully in line with the result in that feature-specific information is temporally persistent—even in this passive setting—far beyond the typical time windows examined in cognitive neuroscience.

Next to the anticipatory and post-sound period, we were also interested in the strength of carrier-frequency specific information following an omission. Since no feedforward activity along the ascending auditory pathway can account for omission-related activity, they have been considered to reflect pure prediction responses[58]. Suggestive evidence comes from an EEG experiment (ref. [36], but see also ref. [37]) in which sounds following button presses with a fix delay could either be random or of a single identity. The authors show evoked omission responses to be sensitive not only to timing, but also to the predicted identity of the stimulus. However, from the evoked response it is not possible to infer which feature-specific information the signal carries (see comment above). Our result significantly extends this research by illustrating that carrier-frequency specific information increases following omission onset the more regular the sound sequence is. Descriptively this occurs rapidly following omission onset with a peak at ~100 ms; however, further peaks can be observed approximately following the stimulation rate until at least 600 ms. The late reactivations of the carrier-frequency specific information of the missing sound is in line with the temporally persisting effects described above, pointing to an enhanced association of past und upcoming stimulus information in the ordered sequence. However, in contrast to the sound centered analysis, the post-omission entropy-related effects are mainly driven by the late training time-window (W2) analogous to the anticipatory effect. Based on this temporal effect, we speculate that while reactivation of carrier-frequency specific information following sounds by further incoming sounds in an ordered sequence engage hierarchically upstream regions in the auditory system, omission-related carrier-frequency specific activity engages downstream areas including some conventionally considered non-auditory (e.g., frontal cortex).

While a predictive processing account appears most parsimonious in explaining the observed findings, one concern could be that adaptation could play a confounding influence. This seems valid at first sight, given the unavoidable increased sequential co-occurence of certain tones in establish a regularity pattern (note that this, however, does not affect the trained classifiers which were all established on the same random condition). For instance, in this study, all transition probabilities across participants were identical and could favor a spread of adaptation to tonotopically neighboring regions[59]. However, a pure adaptation account is unlikely as this process would normally go along with reduced activity that is maximal at the adapted sound frequency and decreases with growing tonotopic distance (for a study in rats, see ref. [60]). Also, an effect of adaptation would be most plausibly seen after the actual sound onset; however, modulations of decoding accuracy across entropy levels was weakest immediately after sound onset or relatively late (see above). Nevertheless, the differential influences of prediction vs adaptation would need to be more systematically tested in the future, perhaps by randomizing the off-diagonal transition probabilities.

The overall individual ability to represent statistical regularities could have profound implications for various behavioral domains[54]. While the main analysis pertained to the decoding of carrier-frequency specific (low-level) information, we also addressed whether a representation of a more abstract feature such as the sequence's entropy level could also be decoded from the noninvasive data. Functionally, extracting regularities requires an integration over a longer time period and previous MEG works focussing on evoked responses have identified in particular slow (DC) shifts as reflecting transitions from random to regular sound[61]. This fits with our result showing that the entropy level of a sound sequence can be decoded above chance at virtually any time point, implying an ongoing (slow) process tracking regularities that is transiently increased following the presentation of a sound. Our across-participant correlation approach is suggestive that indeed the individual's disposition to correctly represent the level of regularity is linked to pre- and post-sound/omission engagement of carrier-frequency specific neural activity patterns. While some open questions remain (e.g., the prestimulus

discrepancy between sound and omission correlation patterns), we consider this line of research very promising for future research studies.

Taken together, the successful decoding of low- and high-level auditory information underlines the significant potential of applying MVPA tools to noninvasive electrophysiological data to address research questions in auditory cognitive neuroscience that would be difficult to pursue using conventional approaches. In particular our approach may be a reliable and easy avenue to parametrize an individual's ability to represent statistical regularities, without the need to invoke behavioral responses that may be influenced by multiple non-specific factors. This could be especially valuable when studying different groups for which conventional paradigms that rely on overt behavior may be problematic, such as children or various patient groups (e.g., disorders of consciousness).

In conclusion, predictive processes should (pre-)engage feature-specific neural assemblies in a top-down manner. However, only little direct evidence exists for this notion in the human auditory system[29,33]. We significantly advance this state by introducing a hybrid regularity modulation and omission paradigm, in which expectations of upcoming carrier-frequency of tones were controlled in a parametric manner. Using MVPA, our results unequivocally show an increase of carrier-frequency specific information during anticipatory as well as (silent) omission periods the more regular the sequence becomes. Our findings and outlined approach holds significant potential to address in-depth the further questions that surround the role of internal models in auditory perception.

## Methods

**Participants**. A total of 34 volunteers (16 females) took part in the experiment, giving written informed consent. At the time of the experiment, the average age was $26.6 \pm 5.6$ SD years. All participants reported no previous neurological or psychiatric disorder, and reported normal or corrected-to-normal vision. One subject was discarded from further analysis, since in a first screening it was found that she had been exposed to a wrong entropy sequence (twice MP and no OR). The experimental protocol was approved by the ethics committee of the University of Salzburg and was carried out in accordance with the Declaration of Helsinki. In particular, written informed consent to take part in this study was obtained from all participants at the beginning of the experiment.

**Stimuli and experimental procedure**. Before entering the MEG cabin, five head position indicator (HPI) coils were applied on the scalp. Anatomical landmarks (nasion and left/right pre-auricular points), the HPI locations, and around 300 headshape points were sampled using a Polhemus FASTTRAK digitizer. After a 5 min resting state session (not reported in this study), the actual experimental paradigm started. The subjects watched a movie ("Cirque du Soleil: Worlds Away") while passively listening to tone sequences. Auditory stimuli were presented binaurally using MEG-compatible pneumatic in-ear headphones (SOUNDPixx, VPixx technologies, Canada). This particular movie was chosen for the absence of speech and dialogue, and the soundtrack was substituted with the sound stimulation sequences. These sequences were composed of four different pure (sinusoidal) tones, ranging from 200 to 2000 Hz, logarithmically spaced (that is: 200 Hz, 431 Hz, 928 Hz, 2000 Hz) each lasting 100 ms (5 ms linear fade in/out). Tones were presented at a rate of 3 Hz. Overall the participants were exposed to four blocks, each containing 4000 stimuli, with every block lasting about 22 min. Each block was balanced with respect to the number of presentations per tone frequency. Within the block, 10% of the stimuli were omitted, thus yielding 400 omission trials (100 per omitted sound frequency). While within each block, the overall number of trials per sound frequency was set to be equal, blocks differed in the order of the tones, which were parametrically modulated in their entropy level using different transition matrices[62]. In more detail, the random condition (RD; see Fig. 1) was characterized by equal transition probability from one sound to another, thereby preventing any possibility of accurately predicting an upcoming stimulus (high entropy). In the ordered condition (OR), presentation of one sound was followed with high (75%) probability by another sound (low entropy). Furthermore, two intermediate conditions were included (midminus and midplus, labeled MM and MP respectively)[62]. The probability on the diagonal was set to be equiprobable (25%) across all entropy conditions, thereby controlling for the influence of self-repetitions. The experiment was programmed in MATLAB 9.1 (The MathWorks, Natick, Massachusetts, U.S.A) using the open source Psychophysics Toolbox[63].

**MEG data acquisition and preprocessing**. The brain magnetic signal was recorded at 1000 Hz (hardware filters: 0.1–330 Hz) in a standard passive magnetically shielded room (AK3b, Vacuumschmelze, Germany) using a whole head MEG (Elekta Neuromag Triux, Elekta Oy, Finland). Signals were captured by 102 magnetometers and 204 orthogonally placed planar gradiometers at 102 different positions. We use a signal space separation algorithm implemented in the Maxfilter program (version 2.2.15) provided by the MEG manufacturer to remove external noise from the MEG signal (mainly 16.6 Hz, and 50 Hz plus harmonics) and realign data to a common standard head position (-trans default Maxfilter parameter) across different blocks based on the measured head position at the beginning of each block[64].

Data analysis was done using the Fieldtrip toolbox[65] (git version 20170919) and in-house built scripts. First, a high-pass filter at 0.1 Hz (6th order zero-phase Butterworth filter) was applied to the continuous data. Subsequently, for independent component analysis, continuous data were chunked in 10 s blocks and down-sampled at 256 Hz. The resulting components were manually scrutinized to identify eye blinks, eye movements, heartbeat and the 16 and 2/3 Hz train power supply artifacts. Finally, the continuous data were segmented from 1000 ms before to 1000 ms after target stimulation onset and the artifactual components projected out ($3.0 \pm 1.5$ SD components removed on average per each subject). The resulting data epochs were down-sampled to 100 Hz for the decoding analysis. Finally, the epoched data was 30 Hz lowpass-filtered (6th order zero-phase Butterworth filter) prior to further analysis. Following these preprocessing steps, no trials were rejected[66].

**Multivariate pattern analysis (MVPA)**. We used MVPA as implemented in the MVPA-Light (https://github.com/treder/MVPA-Light, commit 003a7c) package[67,68], forked and modified in order to extract the classifier weights (https://github.com/gdemarchi/MVPA-Light/tree/devel)[47]. MVPA decoding was performed on single trial sensor-level (102 magnetometers) data using a time-generalization[46] analysis.

Overall, three decoding approaches were taken. At first, in order to investigate how brain activity is modulated by the different entropy levels (*Entropy-level decoding*), we kept trials either with only sound presentation (removing omission trials) or only omissions (discarding the sounds). We defined four decoding targets (classes) based on block type (4 contexts: RD, MM, MP, OR). Sounds that were preceded by a rare (10% of the time) omission were discarded, whereas all the omissions were kept in the omission entropy decoding. Then, to test whether we could classify carrier frequency in general (a.k.a. *Sound-to-sound decoding*), we defined four targets (classes) for the decoding related to the carrier frequency of the sound presented on each trial (4 carrier frequencies). In order to avoid any potential carry-over effect from the previous sound, the classifier was trained only on the random (RD) sounds. Also here the sounds preceded by an omission were discarded. To avoid further imbalance, the number of trials preceding a target sound were equalized, for example, the 928 Hz sound trials were preceded by the same number of 200 Hz, 430 Hz, 928 Hz and 2000 Hz trials (N-1, N balancing). Finally, in order to study whether omission periods contain carrier frequency specific neural activity (*Sound-to-omission decoding*), omission trials were labeled according to the carrier frequency of the sound which would have been presented. As with the sound-to-sound decoding, the random sounds trials were used to train the classifier, which was subsequently applied to a test set of trials where sounds were not presented, that is, omissions, using the same balancing schemes as before.

Using a Multiclass Linear Discriminant Analysis (LDA) classifier, we performed a decoding analysis at each time point around stimulus/omission onset. A five-fold cross-validation scheme, repeated five times, was applied for entropy-level and the random sound-to-sound decoding, whereas for the training on the RD sound—testing on MM, MP, OR sounds as well as for the sound-to-omission decoding—no cross-validation was performed, given the cross decoding nature of the latter testing of the classifier. For the sound-to-omission decoding analysis, the training set was restricted to random sound trials and the testing set contained only omissions. In all cases, training and testing partitions always contained different sets of data.

Classification accuracy for each subject was averaged at the group level and reported to depict the classifier's ability to decode over time (i.e., time-generalization analysis at sensor level). The time generalization method was used to study the ability of each LDA classifier across different time points in the training set to generalize to every time point in the testing set[46]. For the sound-to-sound and sound-to-omissions decoding, time generalization was calculated for each entropy level separately, resulting in four generalization matrices, one for each entropy level. This was necessary to assess whether the contextual sound sequence influences classification accuracy on a systematic level.

**Decoding weights projection analysis**. For relevant time frames the training decoders weights were extracted and projected in the source space as follows. For each participant, realistically shaped, single-shell headmodels[69] were computed by co-registering the participants' headshapes either with their structural MRI (15 participants) or—when no individual MRI was available (18 participants)—with a standard brain from the Montreal Neurological Institute (MNI, Montreal, Canada), warped to the individual headshape. A grid with 1.5 cm resolution based on an MNI template brain was morphed to fit the brain volume of each participant.

LCMV spatial filters were computed starting from the preprocessed data of the training random sounds (for the sound-to-sound and sound-to-omission decoding), or from all the sound or omission data (for the entropy decoding)[70]. Common practice in the field is to multiply the sensor level single-trial time-series by the filter obtained above to create the so-called virtual sensors. Instead, we multiplied the sensor level "corrected by the covariance matrix"[71] decoding weights time-series by the spatials filters to obtain a "informative activity" pattern[47]. Baseline normalization was performed only for visualization purposes (relative change of 100-ms pre-stimulus activity).

**Statistical analysis**. For the sound and omission decoding, we tested the dependence on entropy level using a regression test (*depsamplesregT* in Fieldtrip). Results for sounds and omissions were sorted from random to ordered respectively. In order to account for multiple comparisons, we used a nonparametric cluster permutation test[72] as implemented in Fieldtrip using 10000 permutations and a $p < 0.025$ to threshold the clusters, on a pseudo time-frequency (testing time vs training time—accuracy 2D structure). Moreover, to investigate the temporal dynamics of the single entropy levels in the regression, we ran the statistics above averaging across two different time windows ("avgoverfreq" in fieldtrip), namely an early time window "W1" (from 75 ms to 125 ms post random sound onset in the training set) for the sound-to-sound decoding, and a later window "W2" (from 125 ms to 333 ms).

**Reporting summary**. Further information on research design is available in the Nature Research Reporting Summary linked to this article.

## Data availability
The "mat" files containing the data shown in the figures, along with the MATLAB code to recreate the plots, are available at Zenodo (https://doi.org/10.5281/zenodo.3268713). In the same repository a downsampled (to 100 Hz) version of the raw data is present. The original non resampled data (ca. 300 Gb) is available, upon reasonable request, from the corresponding author.

## Code availability
Data analysis pipeline is available at the corresponding author's github repository (https://github.com//gdemarchi)

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

## Acknowledgements

We thank Dr. Anne Hauswald, Dr. Anne Weise and Mrs. Marta Partyka for helpful comments on earlier versions of the manuscript, and Ms Hayley Prins for proofreading it. We thank Mr. David Opferkuch and Mr. Manfred Seifter for their help with the measurements. For the publication, we acknowledge financial support by the Open Access Publication Fund of the University of Salzburg and by the Center for Cognitive Neuroscience (CCNS) of the University of Salzburg.

## Author contributions

G.D. and N.W. designed the study; G.D. performed the experiments; G.D., G.S., and N.W. designed and performed the analyses; G.D., G.S., and N.W. wrote the manuscript.

## Additional information

**Competing interests:** The authors declare no competing interests.

