## [Peer Review File · Nature Communications]

Reviewers' comments:

Reviewer #1 (Remarks to the Author):

Demarchi et al. recorded MEG data while participants were passively listening to sequences of brief tones. The tone sequences had a varying statistical structure (i.e. entropy), ranging from random (uniform transition probabilities) to ordered (next stimulus 75% predictable). A small subset of tones were omitted. The authors present evidence that evoked responses are present in right primary auditory cortex (A1) for both sounds and omissions, with sounds additionally engaging left A1 and omissions additionally engaging visual cortex. The authors additionally use decoding analyses, the results of which they interpret as evidence that entropy is represented in the cortex, and that tonally specific auditory cortical representations are activated based on predictions about the upcoming stimulus.

I have some reservations with respect to novelty, interpretation, and robustness of the results.

* Novelty: there is already a large body of work out there, showing content-specific activation of sensory templates, also in the auditory domain (e.g. Sanmiguel et al J Neurosci 2013).

* Interpretation: entropy/predictability is manipulated by the degree to which a tone is preceded by itself or a tonal neighbour. Given what we know about tonotopic organization, the results are likely to be confounded by sensory adaptation.

* Robustness: there appear to be some odd analysis choices, and some 'sanity checks' don't seem to show reliable results, which diminishes my confidence in the robustness of the observed findings.

Major:

(1) For the analysis of the effect of regularity on evoked responses, (Figure 2B/C) the normalization introduced for 2C (effect on omissions) appears rather ad hoc. Furthermore, the effect found after normalization localizes to left A1, where no ERF effect was found for omissions; and based on 2C right panel it appears that using this normalization there is no effect of entropy on sound-ERF (an effect present with the analysis in 2B). The fact that two different analysis approaches yield directly conflicting conclusions casts doubt on the reliability of this finding.

(2) The decoding of entropy level is likely confounded by stimulus identity. Although the authors indicate that they balanced class (i.e. entropy level) frequency across train-test splits, there is no information about whether stimulus identities were also balanced across both the train-test splits and entropy levels for this analysis. Furthermore, note that also the identity of stimulus n-1 should be balanced in addition to that of stimulus n (i.e. the authors should balance frequencies of stimulus pairs (n-1, n)), otherwise confounding effects might remain.

(3) I am not convinced that the analysis of residuals after linear regression of classification performance on peristimulus time adequately removes carry-over effects from the previous stimulus (Figure 4C, 5C). The nonlinear "bump" might reflect e.g. the interaction of stimulus n-1 with stimulus n, a pattern which would also be more pronounced in a lower-entropy context. One can imagine many other cortically nonlinear operations which would yield residual decoding accuracy after removing a linear decay. The principled way to approach the problem of carryover is to demonstrate cross-classification: train a decoder for class A only on successive stimuli (B,A) and then test only on pairs (C,A). An additional flaw in the present experimental design is the tight correlation between transition probability and tonal neighbourhood (i.e., tones that are spectrally and thus cortically close also tend to follow one another in the ordered condition); this might make fully orthogonal cross-classification impossible.

Other points, in decreasing order of severity:

(4) The effect reported in line 299 is labelled as “significant albeit at a non-Bonferroni-corrected level”. Given the issues with this approach in general (see point 3), and given that this test is essentially the one crucial test for the main hypothesis of the paper; the reported level of evidence comes across as rather weak.

(5) Beamformer plots are showing clusters masked at 85% in Figure 2, masked at 10% for Figure 3, masked by $p < 0.05$ for Figure 4B, and masked by $p < 0.01$ for Figure 5B. These thresholds appear somewhat ad hoc.

(6) Statistical tests are missing for the main ERF results in Figure 2A.

(7) A very clear feature of the cross-temporal generalization matrices in Figure 4A is the off-diagonal bands for RND, OR, and MP (the latter weakly). It would be good to reflect on these.

(8) Acronyms MM and MP are never defined.

(9) The manuscript occasionally confuses time-generalization analysis (training and testing a classifier on different time points) and ‘standard’ decoding analysis (training and testing on the same time point), e.g. around lines 558-560 and 175-179. Note also that decoding analysis typically does not test for patterns across time; at least not in the sense that a searchlight tests for patterns across space. This equivalence is erroneously suggested in lines 559-560.

(10) To test for dependence of effects on entropy level, the authors used linear regression analysis. This is not really a criticism, but more a suggestion: I wonder whether the sensitivity of these tests might be boosted considerably by using a different analysis. The chosen entropy levels OR, MM, MP, RND, don’t seem to follow each other in a necessarily linear fashion; something akin to an analysis of variance with a categorical independent variable might be more appropriate.

(11) Line 175: were indeed only the magnetometers used? Or also the gradiometers?

(12) Unclear what ‘they’ refers to in line 75.

(13) Typos etc.: tines > tones (line 474); are > is (line 111); replay > preplay (line 54).

Reviewer #2 (Remarks to the Author):

In this review I will focus on methodological and terminological issues.

The major challenge of this research project is how to differentiate whether what is being discovered is prediction-related neural activity, or whether it is an artefact of the analysis: as tones (in the relevant conditions with structure) follow each other in ordered, predictable sequences, the machine classifier can easily pick up on this regularity. The severity is clearly visible in Figures 4 and 5, where the tone can be decoded with high accuracy before it is presented (in the baseline). Consistent with this, the more regular the context, the better the decoding. Same for the significant regularity decoding (Fig. 3) before the stimulus was presented.

The authors acknowledge this problem, and propose a solution based on linear regression, where they regress out previous stimulation carry over activity. The problem with this approach is that decoding accuracy is not linear. The analysis presented here does not allow to distinguish between the main effect and the confound with high confidence.

To remedy this situation, I propose the following course of action. Currently, the authors are

analyzing each regularity/entropy level separately. Because in the high-regularity condition the next tone is predictable from the previous tones, this conflates the main effect of interest with an artefact of the machine analysis. A potential solution is to classify across regularity conditions, in particular from the low-regularity condition to the high-regularity condition (cross-classification). More specifically for the most crucial analysis: train a classifier to distinguish brain data corresponding to tones A vs B (actually played) based on brain data from the no-regularity context, then test the classifier on data from the high-regularity context (the to-be expected tones A and B in silent periods). Because stimulus sequence is random in the no-regularity condition, the classifier cannot pick up on any regularity between tones. If this analysis shows a positive decoding accuracy, this (as far as I can see) undoubtedly would indicate feature-specific predictive representations in silent periods. A secondary analysis from tones in one regularity to another regularity condition should complement this analysis. Third, a comprehensive cross-classification analysis across all levels of regularity should be presented in supplementary material

Data in Figure 3 is subject to the same problem, but I do not see at the moment how it could be solved. As is, it is inconclusive. I suggest removing it from the manuscript, or maybe try a cross-classification across tones. This might establish regularity decoding independent of the previous stimulus. The authors must determine whether this analysis is indeed bias-free.

MINOR POINTS

* I wonder about the phrasing of prediction-related activity being “sharply tuned”. It suggests that the alternative might be broad tuning, or that previous research has only shown broad tuning, while this study shows sharp tuning. What is meant by sharp tuning seems to be that the prediction is of particular contents or features rather than an unspecific signal of prediction per se, so why not call it feature-specific?

* The authors should make more explicit the rationale for choosing time windows 100-200 and 200-300 ms and conducting analysis separately on those windows (Fig 2).

* Caption for Fig. 2 is cut off, which made it difficult to assess the figure in detail

* I do not understand why Fig. 3 only reports the on-diagonal decoding accuracy. First, it is strange to phrase the analysis as a time-generalization analysis and then not show the time-generalization results. In this context I do not understand how the authors can conclude from non-presented data anything about “temporally generalizable neural patterns” (l. 175-176).

* ll. 571/2: were tests one- or two-sided?

Reviewer #3 (Remarks to the Author):

Thank you for asking me to review this stimulating paper. I found it very impressive and generally conducted to an extremely high standard with a high degree of analytical expertise. It will be an important addition to the field, as a comprehensive demonstration of predictive coding for perception. In addition, by demonstrating the ability to decode both the instantiation and reconciliation of predictions in the absence of sensory input, using non-invasive neurophysiological techniques, the authors expand what we think of as possible in for future MEG investigations.

While I think that this is overall an excellent paper, which I would be very pleased to see in print, I do have some specific comments, which I list in order of importance:

1) Much is made of laterality effects, but the source reconstruction method used was LCMV beamformers. These have the property of suppressing correlated information from symmetrical

sources. Therefore, it is entirely unsurprising that the authors found effects that were very strongly side-locked. In all likelihood, this probably represents only a small (and perhaps even chance) difference in information content between right and left auditory cortices, but LCMV attempts to assign such correlated information to a single side. The authors could address this in two ways. They could simply acknowledge this property of their method but justify its use on the basis of the other (many) strengths of LCMV, and note up-front that laterality effects cannot be interpreted. This is fairly unsatisfying in some respects, but does not fundamentally change the authors' primary conclusions. Alternatively, they could replicate their whole analysis using a source reconstruction technique such as minimum norm estimation that does not make laterality assumptions to assess whether their laterality findings remain (NB: they might wish to avoid minimum sparse priors, as this explicitly makes a symmetry assumption). This would be more satisfying, and could either confirm or refute laterality effects, but would clearly be quite a lot of work.

2) I found figure 5A+B beautiful, but I was surprised by the way it was discussed. The best classification accuracies and strongest entropy dependent modulations occur before time zero. This is great, as it demonstrates that the instantiated prediction itself can be decoded with the authors' method (and not just the prediction error signal). However, in the discussion the authors minimise the importance of this finding, providing a potential alternate explanation that I must admit I do not fully understand and which the authors themselves say is not the most parsimonious explanation of the data. At the moment this reads like an acknowledgement of a previous reviewer criticism. I think that the authors need to address this more closely, as the ability to decode the instantiation of predictions would be a major coup and I do not fully understand why they don't feel confident in this.

3) Similarly, in figure 5B the finding that is highlighted is the weaker effect occurring from 200-300ms. This is decoding based on different information to the early cluster, from -100 to 0ms, as there is no overlap of the off-diagonal activity. The authors' interpretation of this cluster is essentially that it represents decoding of the prediction error, and indeed this might well be true. However, the sounds were presented at 3Hz, indicating that this decoding is occurring from -100ms to 0ms relative to the NEXT sound. In the low entropy conditions, the next sound is highly predictable (75%), so might it not be that the authors are in fact decoding the instantiation of the following prediction? Even though this activity follows an omission response, the transition probabilities are still very predictable. Perhaps this could be tested by comparing those 25% of trials where there is stimulus repetition to the 75% where there is not?

4) In figure 4B is it intriguing to see that there is significant left IFG involvement. It would be nice if the authors could discuss this in the context of the MEG literature looking at involvement of left IFG in prediction instantiation and reconciliation, which has mostly been performed in the speech domain. I am thinking especially of Park, *Current Biology* 25.12 (2015): 1649-1653; Sohoglu, *J Neuroscience* 32.25 (2012): 8443-8453; and Cope, *Nature communications* 8.1 (2017): 2154. These papers have all demonstrated univariate and/or connectivity/coherence involvement of frontal regions, so are complementary to the study at hand.

5) In figure 2A the authors claim a difference in peak latency in the evoked responses. This looks self-evidently true from the data, but they should perform some kind of statistical test to back this up. A double dissociation in magnitude by time would suffice, and would probably be better than an attempt to define the peak in individuals.

6) Also figure 2A, why is the source reconstruction only shown for 50-200ms, when the omission response peak is at 300ms. I would have liked to see a second time window of 200-300ms and a condition contrast in each window.

7) In figure 1 legend the acronyms need expansion. They don't appear in-text until the methods.

Response to the reviewers' comments

Reviewer #1 (Remarks to the Author):

R1/1 "Novelty: there is already a large body of work out there, showing content-specific activation of sensory templates, also in the auditory domain (e.g. Sanmiguel et al J Neurosci 2013)."

We tried our best to consider existing works and have added the ones were of particular importance for our study. This included another Sanmiguel et al. paper from the same year (2013) published in Frontiers of Neuroscience. We now also include the Journal of Neuroscience paper. Despite elegant in their design and important to the field we kindly disagree that these works convincingly show content-specific (pre-)activations of sensory templates at least with the same directness that we do in our work (which is even more clear following significant changes to the analysis pipeline). Firstly, these works investigate the statistical regularity of an action to the occurrence of a sound, which is occasionally omitted to investigate the omission response. Especially the Frontiers paper showed that the omission response in this context emerges when a specific prediction about the upcoming sound is formed. However, the association of actions with specific sounds may be to some extent not entirely comparable to the processes that derive statistical regularities and predictions from preceding sound sequences (e.g. in daily life we learn that actions are associated with specific and not random sounds; e.g. speaking or using tools etc). Furthermore, it is impossible to derive feature specific information from the evoked response signals and to what extent this feature specific information is pre-activated (in contrast to decoding approaches) . An important fMRI study that we now add, that appeared following our first submission is also now cited (Berlot et al., 2018, J Neuroscience) that used 7T MRI to uncover tonotopic specific activity patterns during omission periods. Our MEG study supports this finding and goes further to illustrate the precise time-courses (in terms of training- and testing time periods) of prediction-related processes. Importantly, especially following our analysis modifications, our study unequivocally shows the anticipatory dynamics of prediction signals in the auditory system.

R1/2 "Interpretation: entropy/predictability is manipulated by the degree to which a tone is preceded by itself or a tonal neighbour. Given what we know about tonotopic organization, the results are likely to be confounded by sensory adaptation."

We agree that our original analysis carried the confound that a preceding tone in a regular sequence already carries information about the current tone, so that this cannot be fully dissociated from tone-frequency specific information of the current tone (i.e. the one presented at time-point 0) when training and testing within the same condition. This was also noted by Reviewer 2 and we completely modified our analysis approach now. In brief, we only trained our classifiers on random sounds fully circumventing any potential carry-over effects (clearly visible from the flat prestimulus decoding performance in Figure 2A). Thus any prestimulus time-generalization effects observed in Figure 3 can only be attributed to preactivations of carrier-frequency specific information that would be presented at time-point 0. This pattern, including the implication of rather late training-time windows as well as the very different post-stimulus dynamics for sound- and omission-decoding cannot be meaningfully explained by sensory adaptation.

R1/3 "Robustness: there appear to be some odd analysis choices, and some 'sanity checks'"

don't seem to show reliable results, which diminishes my confidence in the robustness of the observed findings."

We completely changed the analysis approach for the current version of the paper to take into account the confounds in our first approach. We have now replicated our results on an independent sample (including tinnitus sufferers; we can informally provide some images upon request; the paper investigating abnormal predictive processing in tinnitus is currently in preparation for a submission elsewhere) and are extremely confident in the main outcomes of this paper.

R1/4: "(1) For the analysis of the effect of regularity on evoked responses, (Figure 2B/C) the normalization introduced for 2C (effect on omissions) appears rather ad hoc. Furthermore, the effect found after normalization localizes to left A1, where no ERF effect was found for omissions; and based on 2C right panel it appears that using this normalization there is no effect of entropy on sound-ERF (an effect present with the analysis in 2B). The fact that two different analysis approaches yield directly conflicting conclusions casts doubt on the reliability of this finding."

Since our main study idea (i.e. studying the pattern of carrier-frequency specific information as a function of statistical regularity) cannot be addressed by studying evoked responses and also falsely suggesting similarities to previous ERP works (e.g. the Sanmiguel et al. ones mentioned above), we decided to remove this part from the manuscript entirely. Our new analysis approach of training classifiers on random sounds and time- and condition-generalizing them drives home the main messages in the clearest form, with other analysis obscuring this clarity.

R1/5: "(2) The decoding of entropy level is likely confounded by stimulus identity. Although the authors indicate that they balanced class (i.e. entropy level) frequency across train-test splits, there is no information about whether stimulus identities were also balanced across both the train-test splits and entropy levels for this analysis. Furthermore, note that also the identity of stimulus $n-1$ should be balanced in addition to that of stimulus n (i.e. the authors should balance frequencies of stimulus pairs $(n-1, n)$), otherwise confounding effects might remain."

We agree with the reviewer. As mentioned in the methods part of the current manuscript, now we are employing this $(n-1, n)$ balancing scheme.

R1/6: "(3) I am not convinced that the analysis of residuals after linear regression of classification performance on peristimulus time adequately removes carry-over effects from the previous stimulus (Figure 4C, 5C). The nonlinear "bump" might reflect e.g. the interaction of stimulus $n-1$ with stimulus n , a pattern which would also be more pronounced in a lower-entropy context. One can imagine many other cortically nonlinear operations which would yield residual decoding accuracy after removing a linear decay. The principled way to approach the problem of carryover is to demonstrate cross-classification: train a decoder for class A only on successive stimuli (B,A) and then test only on pairs (C,A). An additional flaw in the present experimental design is the tight correlation between transition probability and tonal neighbourhood (i.e., tones that are spectrally and thus cortically close also tend to follow one another in the ordered condition); this might make fully orthogonal cross-classification impossible."

We agree that this was a very weak point in our first manuscript. See above for our attempt to circumvent the carryover issue in the present version.

R1/6: “(4) The effect reported in line 299 is labelled as “significant albeit at a non-Bonferroni-corrected level”. Given the issues with this approach in general (see point 3), and given that this test is essentially the one crucial test for the main hypothesis of the paper; the reported level of evidence comes across as rather weak. Find another way to test/look at more efficiently this carry-over effect?”

We agree. See previous response(s).

R1/7: (5) Beamformer plots are showing clusters masked at 85% in Figure 2, masked at 10% for Figure 3, masked by $p < 0.05$ for Figure 4B, and masked by $p < 0.01$ for Figure 5B. These thresholds appear somewhat ad hoc.

We agree. Overall, we think that the “spatial” (source level) pattern is secondary to the feature (carrier-frequency specific) information pattern, which we have made more strong in the current version of the manuscript. Also for deriving spatial “informative activity” patterns we now use an alternative approach analogous to (Marti & Dehaene, *Nat. Commun.* 2017). In total, we try to provide this more as additional information rather than building our main claims on it.

R1/7: “(6) Statistical tests are missing for the main ERF results in Figure 2A.”

ERFs are no longer included in the manuscript.

R1/7: (7) “A very clear feature of the cross-temporal generalization matrices in Figure 4A is the off-diagonal bands for RND, OR, and MP (the latter weakly). It would be good to reflect on these.”

This is an interesting observation that we also noticed. While keeping the entropy decoding (see Figure 4 in the new manuscript), we decided to drop the time-generalization here, in order not to divert too much from the main goal of the manuscript. If the reviewer deems it to be very important we could add this again at least as a supplementary figure.

Off-the-record we actually found this observation so interesting that we have conducted a follow up experiment. According to (Pöppel, *Philos Trans R Soc Lond B Biol Sci.* 2009), among others, the temporal segregation/integration of stimuli can happen at different time scales (“temporal integration windows” (TWs)), that can range from tens of milliseconds up to 2-3 s. In short, in a hopefully soon-to-be-submitted study we manipulated the rate of presentations of the tones (2 vs 3 Hz), using the anti diagonal of the entropy level temporal generalization decoding maps to test this integration windows hypothesis.

R1/8: (8) Acronyms MM and MP are never defined.

This now fixed in the current version of the manuscript.

R1/9: (9) The manuscript occasionally confuses time-generalization analysis (training and testing a classifier on different time points) and ‘standard’ decoding analysis (training and testing on the same time point), e.g. around lines 558-560 and 175-179. Note also that decoding analysis typically does not test for patterns across time; at least not in the sense

that a searchlight tests for patterns across space. This equivalence is erroneously suggested in lines 559-560.

We agree with the reviewer. Now the analysis has been streamlined (e.g. we removed in toto the searchlight analysis in favour of the “informative activity” approach as stated in R1/7), and it’s now fixed in the current version of the manuscript.

R1/10: “(10) To test for dependence of effects on entropy level, the authors used linear regression analysis. This is not really a criticism, but more a suggestion: I wonder whether the sensitivity of these tests might be boosted considerably by using a different analysis. The chosen entropy levels OR, MM, MP, RND, don’t seem to follow each other in a necessarily linear fashion; something akin to an analysis of variance with a categorical independent variable might be more appropriate.”

This is an interesting point, but we are confident that the issues that we introduced this analysis for in the previous version are now circumvented using our new analysis approach.

R1/11: “(11) Line 175: were indeed only the magnetometers used? Or also the gradiometers?”

We used magnetometers only (see lines 262 and 552 of the new manuscript), as other studies did in the past (e.g. Kaiser et al, *J Neurophysiol.* 2016). Based on our pilot measurements and analysis we found that, while getting overall similar patterns when including gradiometers, decoding accuracy was slightly better when taking the magnetometers only. We have not followed this up specifically, but could imagine that large parts of the auditory cortex are not located superficially, could contribute to this circumstance.

R1/12: “(12) Unclear what ‘they’ refers to in line 75.”

This now fixed in the current version of the manuscript.

R1/13: “(13) Typos etc.: tines > tones (line 474); are > is (line 111); replay > preplay (line 54).”

This now fixed in the current version of the manuscript.

Reviewer #2 (Remarks to the Author):

R2/1: “The major challenge of this research project is how to differentiate whether what is being discovered is prediction-related neural activity, or whether it is an artefact of the analysis: as tones (in the relevant conditions with structure) follow each other in ordered, predictable sequences, the machine classifier can easily pick up on this regularity. The severity is clearly visible in Figures 4 and 5, where the tone can be decoded with high accuracy before it is presented (in the baseline). Consistent with this, the more regular the context, the better the decoding. Same for the significant regularity decoding (Fig. 3) before the stimulus was presented.

The authors acknowledge this problem, and propose a solution based on linear regression, where they regress out previous stimulation carry over activity. The problem with this approach is that decoding accuracy is not linear. The analysis presented here does not allow to distinguish between the main effect and the confound with high confidence.

To remedy this situation, I propose the following course of action. Currently, the authors are analyzing each regularity/entropy level separately. Because in the high-regularity condition the next tone is predictable from the previous tones, this conflates the main effect of interest with an artefact of the machine analysis. A potential solution is to classify across regularity conditions, in particular from the low-regularity condition to the high-regularity condition (cross-classification). More specifically for the most crucial analysis: train a classifier to distinguish brain data corresponding to tones A vs B (actually played) based on brain data from the no-regularity context, then test the classifier on data from the high-regularity context (the to-be expected tones A and B in silent periods). Because stimulus sequence is random in the no-regularity condition, the classifier cannot pick up on any regularity between tones. If this analysis shows a positive decoding accuracy, this (as far as I can see) undoubtedly would indicate feature-specific predictive representations in silent periods.

A secondary analysis from tones in one regularity to another regularity condition should complement this analysis. Third, a comprehensive cross-classification analysis across all levels of regularity should be presented in supplementary material

Data in Figure 3 is subject to the same problem, but I do not see at the moment how it could be solved. As is, it is inconclusive. I suggest removing it from the manuscript, or maybe try a cross-classification across tones. This might establish regularity decoding independent of the previous stimulus. The authors must determine whether this analysis is indeed bias-free.”

We thank the Reviewer for this constructive feedback, putting the finger directly on the sensitive issue. Upon rethinking our study we completely agree with this criticism (also raised by Reviewer 1) and we followed the suggestion to completely change the analysis approach. Most importantly we only use the classifier trained on sounds in the random condition to then use in a time- and condition-generalized manner. The flat prestimulus decoding response in Figure 2A indicates that no preceding patterns can be used, i.e. any above chance decoding performance in prestimulus periods in Figure 3 during regular sound sequences has to refer to carrier-frequency specific information expected at time-point 0. Another point that makes us confident that our analysis is bias-free, especially for the anticipatory and omission parts, is the virtual absence of on-diagonal effects in the time- and condition-generalized analysis in Figure 3. Biases would necessarily have to be seen most strongly on-diagonal (hence our clumsy first linear regression approach).

R2/2: “I wonder about the phrasing of prediction-related activity being “sharply tuned”. It suggests that the alternative might be broad tuning, or that previous research has only shown broad tuning, while this study shows sharp tuning. What is meant by sharp tuning seems to be that the prediction is of particular contents or features rather than an unspecific signal of prediction per se, so why not call it feature-specific?”

We agree and we removed the term from the title and put an emphasis more on stating “carrier-frequency specific information”. We kept the term “sharp tuning” at some points since we find it to be a good metaphor to grasp the main finding. However if the Reviewer thinks the term should be abolished entirely, we could revise accordingly.

R2/3: "The authors should make more explicit the rationale for choosing time windows 100-200 and 200-300 ms and conducting analysis separately on those windows (Fig 2)."

The analysis pipeline has been completely changed. Since we do not think that the evoked response is very meaningful for the claims that we are trying to make and even could lead to some confusion regarding to the novelty of the findings (see R1/1), we decided to remove the ERF analysis entirely.

R2/4: "Caption for Fig. 2 is cut off, which made it difficult to assess the figure in detail"

We apologize for this inconvenience (likely introduced while converting the doc to a pdf) that skipped our attention. Based on our new analysis approach all Figures have changed.

R2/5: "I do not understand why Fig. 3 only reports the on-diagonal decoding accuracy. First, it is strange to phrase the analysis as a time-generalization analysis and then not show the time-generalization results. In this context I do not understand how the authors can conclude from non-presented data anything about "temporally generalizable neural patterns" (l. 175-176)."

We agree. We now (see Figure 4) only have included the on-diagonal result, with the omission data added. There are some interesting patterns in the time-generalization outcomes that have not been included in the main manuscript, as they divert from the main question. We are addressing some of these issues (in particular to what extent off-diagonal decoding patterns could be an indicator integration time-windows for extracting statistical regularities) in a separate experiment (see R1/7).

R2/6: "ll. 571/2: were tests one- or two-sided?"

ERFs are no longer part of the current manuscript (see R1/4).

Reviewer #3 (Remarks to the Author):

R3/1: "1) Much is made of laterality effects, but the source reconstruction method used was LCMV beamformers. These have the property of suppressing correlated information from symmetrical sources. Therefore, it is entirely unsurprising that the authors found effects that were very strongly side-locked. In all likelihood, this probably represents only a small (and perhaps even chance) difference in information content between right and left auditory cortices, but LCMV attempts to assign such correlated information to a single side. The authors could address this in two ways. They could simply acknowledge this property of their method but justify its use on the basis of the other (many) strengths of LCMV, and note up-front that laterality effects cannot be interpreted. This is fairly unsatisfying in some respects, but does not fundamentally change the authors' primary conclusions. Alternatively, they could replicate their whole analysis using a source reconstruction technique such as minimum norm estimation that does not make laterality assumptions to assess whether their laterality findings remain (NB: they might wish to avoid minimum sparse priors, as this explicitly makes a symmetry assumption). This would be more satisfying, and could either confirm or refute laterality effects, but would clearly be quite a lot of work."

We thank the Reviewer for the positive assessment.

Regarding the laterality effect, we wanted to firstly point out that we think that the primary contribution of the paper is to illustrate the temporal dynamics of prediction related feature-specific neural patterns, rather than shedding light on precise anatomical regions (i.e. focus “what” rather than “where”). Overall we tried to down-tone any parts that appeared too strong in this respect. This is partially also a reason why we dropped the searchlight approach in the current manuscript and opted for an approach forwarded by Marti & Dehaene (Marti & Dehaene, *Nat. Commun.* 2017) making suggestive statements about potentially contributing regions. Furthermore, our experience with LCMV is very different to the issue that the Reviewer raises, i.e. this method is very able in separating left and auditory cortices (see Todorovic et al. *J Neurosci.* 2011; see also Figure 2 in original submission, which has been now dropped; see e.g. <http://www.fieldtriptoolbox.org/tutorial/salzburg/> for a practical implementation). The issue of correlated sources is particularly pertinent when correlations are very strong, which is usually not the case when basing the covariance estimation on the single trials (rather than e.g. on the evoked response itself).

R3/1: “2) I found figure 5A+B beautiful, but I was surprised by the way it was discussed. The best classification accuracies and strongest entropy dependent modulations occur before time zero. This is great, as it demonstrates that the instantiated prediction itself can be decoded with the authors’ method (and not just the prediction error signal). However, in the discussion the authors minimise the importance of this finding, providing a potential alternate explanation that I must admit I do not fully understand and which the authors themselves say is not the most parsimonious explanation of the data. At the moment this reads like an acknowledgement of a previous reviewer criticism. I think that the authors need to address this more closely, as the ability to decode the instantiation of predictions would be a major coup and I do not fully understand why they don’t feel confident in this.”

Thank you for the encouraging feedback. The issue pertains to training and testing on the same condition (albeit from omission to sounds): A sound in a regular sequence always contains information about the upcoming sound. So if we are interested in the carrier-frequency specific of an upcoming sound especially in an anticipatory period, this will necessarily generate a confound (i.e. is it information related to the previous sound frequency or patterns related to the upcoming sound frequency). We tried to address using the linear regression approach, which however is not really a good fix for this issue. In the current version of the manuscript we circumvent this issue (see also R2/1) by training our classifier on random sounds and time- and condition generalizing them. Via this approach we removed this confound and made the effects (especially the anticipatory ones) even stronger (see new Figure 3).

R3/2: “3) Similarly, in figure 5B the finding that is highlighted is the weaker effect occurring from 200-300ms. This is decoding based on different information to the early cluster, from -100 to 0ms, as there is no overlap of the off-diagonal activity. The authors’ interpretation of this cluster is essentially that it represents decoding of the prediction error, and indeed this might well be true. However, the sounds were presented at 3Hz, indicating that this decoding is occurring from -100ms to 0ms relative to the NEXT sound. In the low entropy conditions, the next sound is highly predictable (75%), so might it not be that the authors are in fact decoding the instantiation of the following prediction? Even though this activity follows

an omission response, the transition probabilities are still very predictable. Perhaps this could be tested by comparing those 25% of trials where there is stimulus repetition to the 75% where there is not?"

We agree. See comment above.

R3/3: "4) In figure 4B is it intriguing to see that there is significant left IFG involvement. It would be nice if the authors could discuss this in the context of the MEG literature looking at involvement of left IFG in prediction instantiation and reconciliation, which has mostly been performed in the speech domain. I am thinking especially of Park, Current Biology 25.12 (2015): 1649-1653; Sohoglu, J Neuroscience 32.25 (2012): 8443-8453; and Cope, Nature communications 8.1 (2017): 2154. These papers have all demonstrated univariate and/or connectivity/coherence involvement of frontal regions, so are complementary to the study at hand."

See our comment above, that we are trying to reduce the focus on the "where" to focus more on the "what". The analysis former Figure 4B is obtained from the searchlight analysis that still contains the confounding carry-over effect. While it does point to the involvement of this region in prediction processes, we abandoned the within-condition training and testing approach so that we can more confidently bring across the anticipatory effects. I.e. the main source effects are the informative activity derived from random sound presentation (see Figure 2). We do see some bilateral spread to frontal regions in later intervals that are in particular capturing our (anticipatory) prediction effects (see Figure 3), but we are not comfortable in making too strong anatomical claims based on our present analysis.

R3/4: "5) In figure 2A the authors claim a difference in peak latency in the evoked responses. This looks self-evidently true from the data, but they should perform some kind of statistical test to back this up. A double dissociation in magnitude by time would suffice, and would probably be better than an attempt to define the peak in individuals."

The analysis pipeline has been completely changed. Since we do not think that the evoked response is very meaningful for the claims that we are trying to make and even could lead to some confusion regarding to the novelty of the findings (see R1/1), we decided to remove the ERF analysis entirely.

R3/5: 6) Also figure 2A, why is the source reconstruction only shown for 50-200ms, when the omission response peak is at 300ms. I would have liked to see a second time window of 200-300ms and a condition contrast in each window.

See previous comment.

R3/6: "7) In figure 1 legend the acronyms need expansion. They don't appear in-text until the methods."

This now fixed in the current version of the manuscript.

Reviewers' comments:

Reviewer #1 (Remarks to the Author):

The authors have made several improvements to the paper, and it is in much better shape. My major concern remains, which cannot be solved by a different analytical approach. The concern is this: "entropy/predictability is manipulated by the degree to which a tone is preceded by itself or a tonal neighbour. Given what we know about tonotopic organization, the results are likely to be confounded by sensory adaptation."

The authors aimed to solve it by changing their analytical approach. But this is an issue of experimental design, not of analysis choice. There is, in my opinion, simply no way to distinguish between adaptation and entropy/predictability, given the way the study was designed.

Therefore, I think it would be important to clearly mention this caveat in the discussion, so that readers are aware of it and can interpret the data in the light of this limitation.

Reviewer #2 (Remarks to the Author):

As in my previous review I will limit myself to methodological topics. The authors have taken up the suggestion of directed cross-classify from random to non-random sequences in order to establish predictive and content-specific activity in MEG. The results were positive. This has eliminated my worries and I now feel their conclusions are warranted by strong and convincing evidence.

I have three comments in descending order of importance:

1) Only regression results of decoding accuracy vs. level are shown, but not the raw decoding data that goes into the analysis. While this form to summarize data is appropriate, readers would strongly benefit from seeing on what evidence basis the main analysis rests.

2) ll 17-19 (abstract): this sentence will be hard to grasp for naïve readers in its meaning and relevance

3) l 143 The authors use Bonferroni correction – what are the multiple comparisons that are corrected for?

Reviewer #3 (Remarks to the Author):

Thank you for asking me to re-review this paper, which I enjoyed on first viewing. The revised paper is much changed, and significantly pared-down. This renders it more focussed and robust, but also narrower. However, the core result on which the authors have focussed, namely that sharply-tuned auditory predictions can be decoded even in the absence of sound presentation, remains interesting and exciting.

I am still positive about this study, which I think is an important contribution to the field, providing evidence in keeping with the hypotheses of predictive coding, and expanding what we think of as possible with MVPA for MEG. However, as the analysis has been completely reframed, I have some

new comments and concerns, which all-told are rather minor.

I found figure 3 hard to comprehend. I had to read the legend about four times before I think I got it. The eye is immediately drawn to the large, early decoding accuracy peak in lower panel 3A, but I understand that this is not the feature of interest, as it persists across conditions and largely simply represents decoding of sound identity. Am I correct in my understanding that the upper panel t-values are between-condition differences, and in lower panel 3A the eye is supposed to be drawn to the late separation of the curves? This is hard to see, because of axis scaling, and requires a closeness of examination that I think most readers will lack. I also don't really understand why the authors show only the W1 results for sound onset, as it looks like there are also statistically significant between-condition differences in the W2 window for this condition.

I am not particularly convinced by the description of the decoding accuracy in figure 3B represents the 3Hz carrier signal. 90, 330 and 580ms are separated by 240-250ms, which is rather a 4Hz frequency. This seems quite a sizeable discrepancy, given how predictable and consistent the sequence presentation frequency was.

Figure 4A – could the authors statistically test for a difference in accuracy between sound and omission trials at each timepoint?

It wasn't clear to me how the cluster statistics were performed for the correlation analyses in figure 4B, and this is crucial because the results only just pass the threshold for significance. Were cluster permutations in Fieldtrip also used, as for the simpler analyses?

I still don't think that LCMV beamformers are the best source reconstruction method for a multi-voxel analysis of this type, but the authors have dropped their stronger claims about laterality so I no longer think that this is particularly problematic.

Minor points:

Figure 2A is not referenced in the text – although it's pretty clear that this corresponds to approx. line 142

Figure 3A upper – the 0.6s tick mark is missing

Figure 4 – I think it would be helpful to mark on the onset time of the next sound, to remind the skim-reading reader that the second peak is due to a new sound, given the 3Hz presentation rate.

Figure 4, inset – how is informative activity calculated? What is the scale? I see that the authors have correctly regressed out shared covariance (Haufe 2014, Neuroimage), so these projections are robust, but the legend should contain more information about how a naïve reader should interpret the scaling and thresholding. It's also slightly misleading to link this to the peak, given that the analysis was performed over the whole 0-330ms time window.

At times the English is clumsy, and would benefit from proof-reading by a native speaker. E.g.:
"Building upon this integrated feedforward and topdown architecture, cortical and subcortical regions seem to be involved towards auditory predictionerror generation mechanisms."

Sometimes this impairs understanding E.g.:

"An interesting follow-up question, is whether interindividual variability to derive the statistical regularity from the sound sequences would be correlated with carrier frequency specific information for predicted sounds"

Lines 330 and 342 there are open parentheses that are unmatched.

Response to the reviewers' comments

Reviewer #1 (Remarks to the Author):

R1/1 *"The authors have made several improvements to the paper, and it is in much better shape. My major concern remains, which cannot be solved by a different analytical approach. The concern is this: "entropy/predictability is manipulated by the degree to which a tone is preceded by itself or a tonal neighbour. Given what we know about tonotopic organization, the results are likely to be confounded by sensory adaptation."*

The authors aimed to solve it by changing their analytical approach. But this is an issue of experimental design, not of analysis choice. There is, in my opinion, simply no way to distinguish between adaptation and entropy/predictability, given the way the study was designed.

Therefore, I think it would be important to clearly mention this caveat in the discussion, so that readers are aware of it and can interpret the data in the light of this limitation"

We agree that the dissociation between prediction (top-down driven) and adaptation (bottom-up driven) is not trivial given the nature of the paradigm. Also we did not mean to have conclusively solved it by our new analysis strategy. What we meant was that this confound of having a sound systematically preceded by a (tonotopically) neighbouring sound was effectively abolished by *training* the classifier on random sounds sequences. Since the same classifier was used consistently, this leaves open the issue whether adaptation effects could have influenced our results when *testing* the classifier for the separate entropy levels. Note that these could only be caused by a tonal neighbour, as the rate of self-repetitions was strictly equal between the entropy conditions. While this may be in principle possible, we think that an adaptation account (at least in a very strict sense) is not the most parsimonious explanation for our results. Firstly, adaptation would normally go along with reduced activity that is maximal at the adapted frequency and decreases with growing distance (for a rat work see (Dezfouli and Daliri, 2015)). Even though one should be careful to not equate "activity" with "information", sensory adaptation does not make an increase in decoding accuracy with decreasing entropy very likely. Secondly, extending the previous thought, it would be puzzling why effects shown in **Figure 3** are only present prior to stimulus onset (**C** and **D**) and during omissions (**D**), but not when the actual sound was presented. However we acknowledge that this issue would need to be more systematically tested in future, perhaps by randomizing the off-diagonal transition probabilities across participants in future experiments.

To raise the reader's awareness for this issue, we have added the following sentences in the discussion (p. 22 ll. 466-479):

"While a predictive processing account appears most parsimonious in explaining the observed findings, one concern could be that adaptation could play a confounding influence. This seems valid at first sight, given the unavoidable increased sequential co-occurrence of certain tones in establish a regularity pattern (note that this, however, does not affect the trained classifiers which were all established on the same random condition). For instance, in this study, all transition probabilities across participants were identical and could favor a spread of adaptation to tonotopically neighbouring regions (Herrmann et al., 2014). However, a pure adaptation account is unlikely as this process would normally go along with reduced activity that is maximal at the adapted sound frequency and decreases with growing tonotopic distance (for a study in rats, see (Dezfouli and Daliri, 2015)). Also, an effect of

adaptation would be most plausibly seen after the actual sound onset; however, modulations of decoding accuracy across entropy levels was weakest immediately after sound onset or relatively late (see above). Nevertheless, the differential influences of prediction vs adaptation would need to be more systematically tested in the future, perhaps by randomizing the off-diagonal transition probabilities.”

Reviewer #2 (Remarks to the Author):

“As in my previous review I will limit myself to methodological topics. The authors have taken up the suggestion of directed cross-classify from random to non-random sequences in order to establish predictive and content-specific activity in MEG. The results were positive. This has eliminated my worries and I now feel their conclusions are warranted by strong and convincing evidence.”

We want firstly to thank Reviewer #2 for the overall positive assessment and for the critical suggestion on cross classify using the random sequences only. This has made both the analysis and discussion leaner and more straightforward. On the specific points:

R2/1: “Only regression results of decoding accuracy vs. level are shown, but not the raw decoding data that goes into the analysis. While this form to summarize data is appropriate, readers would strongly benefit from seeing on what evidence basis the main analysis rests.”

This comment and comments made by Reviewer #3 made us realize that it is indeed difficult to understand what is actually shown in **Figure 3** without making too many excursions into the legend and main text. Since this figure is carrying the core message of the entire manuscript, we picked up your suggestion and added the “raw” (grand averaged) time-generalization plots on which the regression analysis was based. We also now added the line plots for both (data-driven) time-windows of interest to the sound and omission condition for the sake of consistency. Adding this information carries the danger of making the figure too crowded, but since this is the main figure of the paper, we didn’t want to hide important details in the (often neglected) Supplementary Materials. Thus, we hope that the way we organized the figure drives home the central points of the results.

R2/2: “ll 17-19 (abstract): this sentence will be hard to grasp for naïve readers in its meaning and relevance”

Thank you for pointing out this issue. We agree that the original formulation may be rather technical for the non-specialist reader. The current version contains a completely rewritten and hopefully greatly facilitated abstract in which we attempted to drop all technical terms, that do not serve in bringing across the main points of the study.

The current version of the abstract is now formulated in the following manner (p. 1, ll. 8-20):
“Prior experience enables the formation of expectations of upcoming sensory events. It is not well established in the auditory modality if prediction-related neural signals carry feature-specific information. In this study, we looked at whether auditory predictions are tuned to even carry tonotopic specific information. Participants passively listened to sound sequences of four carrier frequencies with a fixed stimulation rate, ensuring strong temporal expectations of *when* a stimulus would occur. However, feature expectation of *what* stimulus would occur was parametrically modulated across the sequences and sounds were

occasionally omitted. Classifiers were trained to decode the carrier frequencies over time and were subsequently tested for all sounds and occasional omissions across all conditions. Exploiting the excellent temporal resolution of Magnetoencephalography (MEG), we show that increasing the regularity of the sequence boosts carrier-frequency specific neural activity patterns during the anticipatory and omission periods. Our results illustrate that even without bottom-up input, auditory predictions can activate tonotopically specific templates.”

R2/3: “1143 The authors use Bonferroni correction – what are the multiple comparisons that are corrected for?”

We did a t-test against chance for each time point (hence the multiple comparison). Control for multiple comparisons were thus made by setting the critical p value to $.05 / 101 \sim 5 \times 10^{-4}$ (101 being the number of time points between -.3s and .7s, at 100 Hz sampling rate) for **Figure 2A** and to $.05 / 81 \sim 6 \times 10^{-4}$ (# of time points between -.1s and .7s) for **Figure 4A**.

Reviewer #3 (Remarks to the Author):

“Thank you for asking me to re-review this paper, which I enjoyed on first viewing. The revised paper is much changed, and significantly pared-down. This renders it more focussed and robust, but also narrower. However, the core result on which the authors have focussed, namely that sharply-tuned auditory predictions can be decoded even in the absence of sound presentation, remains interesting and exciting.”

We thank the reviewer for the (renewed) enthusiastic assessment.

“I am still positive about this study, which I think is an important contribution to the field, providing evidence in keeping with the hypotheses of predictive coding, and expanding what we think of as possible with MVPA for MEG. However, as the analysis has been completely reframed, I have some new comments and concerns, which all-told are rather minor.”

We made our best effort to address all the remaining concerns and hope that the modifications are satisfactory for the reviewer.

R3/1: “I found figure 3 hard to comprehend. I had to read the legend about four times before I think I got it. The eye is immediately drawn to the large, early decoding accuracy peak in lower panel 3A, but I understand that this is not the feature of interest, as it persists across conditions and largely simply represents decoding of sound identity. Am I correct in my understanding that the upper panel t-values are between-condition differences, and in lower panel 3A the eye is supposed to be drawn to the late separation of the curves? This is hard to see, because of axis scaling, and requires a closeness of examination that I think most readers will lack. I also don’t really understand why the authors show only the W1 results for sound onset, as it looks like there are also statistically significant between-condition differences in the W2 window for this condition.”

We thank the reviewer for this valuable feedback and apologize that s/he had to reverse engineer what is being displayed. The reviewer is correct with regards to what we intended to display. Combined with some feedback by reviewer #2 we completely redesigned **Figure 3**. Mainly we added the raw time-generalization plots and visual aids (an arrow indicating the

arrangement of plots according to entropy level). We hope that by this arrangement, the statistical results (shown now in **Figure 3C** and **D**) become more intuitive to understand. Finally, we scaled the line plots for the two (data driven) time windows of interest so that the condition effects can be more easily appreciated.

R3/2: "I am not particularly convinced by the description of the decoding accuracy in figure 3B represents the 3Hz carrier signal. 90, 330 and 580ms are separated by 240-250ms, which is rather a 4Hz frequency. This seems quite a sizeable discrepancy, given how predictable and consistent the sequence presentation frequency was".

The reviewer is correct. We deleted "thus tracking the stimulation rate of the sound sequence" from the respective sentence.

R3/3: "Figure 4A – could the authors statistically test for a difference in accuracy between sound and omission trials at each timepoint?".

Now we inserted into the figure the explicit comparison (black line) between the sound- (green line) and omission- (red line) locked entropy decoding curves, and we added following explanation to the main text (p. 15, l. 287-290):

"We also tested for a difference in accuracy between sound and omission trials at each timepoint, which was significant ($p < .05$, Bonferroni corrected, black horizontal line) between ~30 ms and ~300 ms, again emphasizing the brevity of this effect."

R3/4: "It wasn't clear to me how the cluster statistics were performed for the correlation analyses in figure 4B, and this is crucial because the results only just pass the threshold for significance. Were cluster permutations in Fieldtrip also used, as for the simpler analyses?".

Yes, statistical significance was determined using nonparametric cluster permutation testing as implemented in fieldtrip. This is now made more specific whenever appropriate. Also we point out the explorative (ad-hoc) nature of this analysis more clearly now at different points in the Results (see p.16, l. 312-316) and Discussion (see p. 23 l. 494-497). We try not to be definitive based on the mixed result as correctly noted by the reviewer, but find it sufficiently interesting to (hopefully) inspire some follow-up research.

R3/5: "I still don't think that LCMV beamformers are the best source reconstruction method for a multi-voxel analysis of this type, but the authors have dropped their stronger claims about laterality so I no longer think that this is particularly problematic.".

We are aware that the interpretation of M/EEG classifier weights in a spatial sense is a non-trivial issue even at the sensor level and there are no generally accepted best practices to follow as e.g. in the fMRI domain. In the original submission we used a searchlight analysis, but have now adapted an alternative strategy as first proposed by (Marti and Dehaene, 2017) that projects (covariance-corrected) classifier weights into source space. We have had extremely positive experience with this approach lately, showing e.g. localization of "informative activity" to right FFA when bistable images are interpreted as face vs house respectively (Rassi et al., 2019) or localization of "informative activity" to left STG when auditory stimuli were classified whether they were part of a memory set or not (Kraft et al., 2019). The source images are mainly shown for descriptive purposes and -as the reviewer correctly notes- we do not intend to build big claims upon them. If the reviewer

agrees, we prefer not to add a lengthy discussion on this issue to the manuscript in order to keep it streamlined.

Minor points:

Figure 2A is not referenced in the text – although it's pretty clear that this corresponds to approx. line 142

Thank you for pointing this out. We have added the reference accordingly.

Figure 3A upper – the 0.6s tick mark is missing

Thanks for catching it. Now we've recreated the entire figure, hopefully without gross mistakes.

Figure 4 – I think it would be helpful to mark on the onset time of the next sound, to remind the skim-reading reader that the second peak is due to a new sound, given the 3Hz presentation rate.

Many thanks for the hint. Now we added the next sound(s) onset both to **Figure 4**, as well as to **Figure 3**.

Figure 4, inset – how is informative activity calculated? What is the scale? I see that the authors have correctly regressed out shared covariance (Haufe 2014, Neuroimage), so these projections are robust, but the legend should contain more information about how a naïve reader should interpret the scaling and thresholding. It's also slightly misleading to link this to the peak, given that the analysis was performed over the whole 0-330ms time window.

Thank you for raising this point. Now we redid **Figure 4**, and a grey horizontal line encompasses the the entire period (0-330 ms) that was used to compute the informative activity brain plot. We also fixed all the legends of **Figure 2** and **Figure 4**, where brains are shown, stating clearly that a relative change baseline has been computed on the (-100-0 ms) time interval, and an arbitrary threshold of 50% was chosen for display purposes.

At times the English is clumsy, and would benefit from proof-reading by a native speaker.

E.g.:

“Building upon this integrated feedforward and topdown architecture, cortical and subcortical regions seem to be involved towards auditory predictionerror generation mechanisms.”

Sometimes this impairs understanding E.g.:

“An interesting follow-up question, is whether interindividual variability to derive the statistical regularity from the sound sequences would be correlated with carrier frequency specific information for predicted sounds”

The original submission went duly through a proof-reader, but for the resubmission our proof-reader of trust was unavailable, therefore quickly before the submission we resorted to trust solely our (non native) English knowledge. Wrongly. Now it's fixed, thanks.

Lines 330 and 342 there are open parentheses that are unmatched.

Thanks. We have fixed the issue now.

References

- Dezfouli MP, Daliri MR. 2015. The Effect of Adaptation on the Tuning Curves of Rat Auditory Cortex. *PLOS ONE* **10**:e0115621. doi:10/f68h5g
- Herrmann B, Schlichting N, Obleser J. 2014. Dynamic Range Adaptation to Spectral Stimulus Statistics in Human Auditory Cortex. *J Neurosci* **34**:327–331. doi:10/f5m4k4
- Kraft N, Demarchi G, Weisz N. 2019. Auditory cortical alpha desynchronization prioritizes the representation of memory items during a retention period. *bioRxiv* 626929. doi:10.1101/626929
- Marti S, Dehaene S. 2017. Discrete and continuous mechanisms of temporal selection in rapid visual streams. *Nat Commun* **8**. doi:10.1038/s41467-017-02079-x
- Rassi E, Wutz A, Müller-Voggel N, Weisz N. 2019. Pre-stimulus feedback connectivity biases the content of visual experiences. *bioRxiv* 437152. doi:10/gf2q4z

REVIEWERS' COMMENTS:

Reviewer #3 (Remarks to the Author):

This re-revision has addressed my remaining concerns, and the new figure 3 much clearer. I look forward to seeing the article in print.

Dr Thomas E Cope